# 1.5 °C degrowth scenarios suggest the need for new mitigation pathways

Lorenz T. Keyßer [1,2 ✉] & Manfred Lenzen [1]

1.5 °C scenarios reported by the Intergovernmental Panel on Climate Change (IPCC) rely on combinations of controversial negative emissions and unprecedented technological change, while assuming continued growth in gross domestic product (GDP). Thus far, the integrated assessment modelling community and the IPCC have neglected to consider degrowth scenarios, where economic output declines due to stringent climate mitigation. Hence, their potential to avoid reliance on negative emissions and speculative rates of technological change remains unexplored. As a first step to address this gap, this paper compares 1.5 °C degrowth scenarios with IPCC archetype scenarios, using a simplified quantitative representation of the fuel-energy-emissions nexus. Here we find that the degrowth scenarios minimize many key risks for feasibility and sustainability compared to technology-driven pathways, such as the reliance on high energy-GDP decoupling, large-scale carbon dioxide removal and large-scale and high-speed renewable energy transformation. However, substantial challenges remain regarding political feasibility. Nevertheless, degrowth pathways should be thoroughly considered.

[1] ISA, School of Physics A28, The University of Sydney, Sydney, NSW, Australia. [2] Department of Environmental Systems Science, Institute for Environmental Decisions, ETH Zürich, Zürich, Switzerland. ✉email: lkeysser@student.ethz.ch

Five years after the Paris Agreement, $CO_2$ emissions are still rising[1], and mitigation timelines for the 1.5 °C and 2 °C climate target become ever more stringent[2]. Meanwhile, integrated assessment model (IAM) mitigation scenarios reported by the Intergovernmental Panel on Climate Change (IPCC) Special Report on 1.5 °C (SR1.5) rely on controversial amounts of carbon dioxide removal and/or on unprecedented technological changes[2,3]. Simultaneously, all of them assume continued growth in gross domestic product (GDP), among other reasons because this is deemed necessary to support societal wellbeing[4]. However, continued GDP growth is widely associated with increasing mitigation challenges, e.g., with increasing energy and material consumption[5–8]. In contrast, alternative mitigation pathways as examined by the expanding degrowth literature[9], are almost completely neglected by the IAM community and the IPCC[2,4]. Thus, their potential to avoid negative emissions and technological change remains unexplored. In this paper, we present an in-depth comparison of IPCC IAM and degrowth mitigation scenarios by applying a simplified quantitative model of the fuel-energy-emissions nexus.

IAMs are widely used for examining interlinkages between social and biophysical systems and their scenarios are prominently applied within climate change mitigation research[2]. These scenarios are based on different sets of assumptions about future population and economic growth, income distribution, as well as behavioural and technological change. The five shared socio-economic pathways are an influential set of such scenarios[2]. Moreover, the IPCC SR1.5[2] includes other scenarios such as the low energy demand (LED) scenario by Grubler et al.[3], which minimises the need for carbon dioxide removal by substantially increased energy and material efficiency, thus strongly decoupling energy and material use from GDP growth. In order to meet the 1.5 °C target, unprecedented transformations of energy, land, infrastructure and industrial systems are necessary[2]. The technological transformation is especially extraordinary for negative emission technologies (NETs), as all scenarios assessed in the IPCC SR1.5 assume carbon dioxide removal of between 100 and 1000 billion tonnes of $CO_2$ ($GtCO_2$) until 2100, mostly through bioenergy to carbon capture and storage (BECCS), and to a lesser extent through afforestation and reforestation (AR). However, the large-scale NETs deployment of several hundred $GtCO_2$ faces substantial uncertainty as well as sustainability and feasibility concerns[10–13].

None of the 222 scenarios in the IPCC SR1.5 and none of the shared socioeconomic pathways projects a declining GDP trajectory[2,4], as is examined by the expanding degrowth literature[9]. Interestingly, empirical evidence[5–8,14] corroborates the degrowth hypothesis[9] that there is a stronger than commonly recognised relationship between the growth in GDP and energy, material and fossil fuel use. Consequently, measures to drastically reduce the latter would also reduce GDP growth[5–7,9,14,15]. A GDP reduction is thus not an end in itself, but embraced as a likely outcome of the necessary ecological and social measures. Degrowth is hence defined as (p. 7) 'equitable downscaling of throughput [that is the energy and resource flows through an economy, strongly coupled to GDP], with a concomitant securing of wellbeing'[9]. On wellbeing, research[16,17] shows that high-income countries could scale back their biophysical impact (and GDP), while maintaining or even increasing[9,18] social performance and achieving higher equity among countries. Thus, intra- and intergenerational equity aspects can be taken into account[9,17,19], e.g., by making the world economy structurally fairer and redistributing from global North to South[17,18]. Further, bottom–up studies show that high living standards can be maintained with substantially less per capita energy use than currently consumed in affluent countries[20]. However, to ensure that such reductions do not lead to the socially harmful and inequitable effects of a recession requires deep socioeconomic changes and policy reforms, such as universal basic services, maximum incomes, working time reductions, and democratic firm ownership[9,17,19,21]. Degrowth scenarios have been explored for single countries[22] and only recently globally with complex IAMs[23].

As a first step to address the lack of IAM-based climate change mitigation scenarios describing degrowth and to encourage further research in this area, this article assesses how degrowth scenarios perform compared with IPCC SR1.5 IAM scenario archetypes regarding key relative risk indicators for feasibility and sustainability. We define feasibility, following the IPCC[2] (p. 52), as 'the capacity of a system as a whole to achieve a specific outcome', in our case, a scenario. We additionally distinguish between socio-technical feasibility, broadly following Loftus et al.[24] (i.e., energy-GDP decoupling, speed and scale of the renewable energy transition and NETs deployment), as well as socio-political feasibility, which includes economic feasibility, broadly following Jewell & Cherp[25]. The latter defines an outcome as politically feasible (p. 2) 'if there is an agent or group of agents who have the capacity to carry out a set of actions which will lead to that outcome in a given context.' To conduct this analysis, we apply a simplified quantitative model of the global fuel-energy-emissions nexus, arriving at several climate change mitigation scenarios and their indicator values. This modelling approach is chosen to complement complex IAMs by enhancing transparency and understanding[26] as well as avoiding common limitations of IAMs, especially regarding degrowth modelling (see Methods and Discussion). A full version of our model can be accessed in Supplementary Data 1. We then assess the relative performance of our scenarios concerning the modelled risk indicators for socio-technical feasibility and sustainability, equity as well as socio-political feasibility. Our results indicate that degrowth scenarios minimise many key risks for feasibility and sustainability, but substantial challenges remain regarding political feasibility. At last, we discuss limitations of our indicators and modelling approach as well as the implications for future research and modelling of the IAM and climate mitigation communities. Here, we conclude that future modelling research should thoroughly consider degrowth scenarios.

## Results

In this section, we firstly describe the scenarios modelled with our simplified representation. Then, we summarise the scenario results. At last, we show how our scenarios perform relative to our indicators, reviewing literature on the significance and interpretation of energy-GDP decoupling, speed and scale of the renewable energy transition, NETs, equity as well as socio-political feasibility.

**Scenario overview**. We investigate the following:

- four pathways with low energy-GDP decoupling (the consumption-driven degrowth pathways: 'Degrowth', 'Degrowth-FullNETs', 'Degrowth-NoNNE' and 'DLE' (Decent Living Energy[20])),
- ten scenarios with medium energy-GDP decoupling (the technology-driven scenarios: 'Moderate', 'Moderate-FullNETs', 'Strong', 'Extreme', 'Utopian', 'IPCC', 'IPCC-FullNETs', 'IPCC-NoNNE', 'ClimateAnalytics' and 'Dec-Moderate'),
- as well as four technology-driven pathways with high energy-GDP decoupling (called 'Dec-Strong', 'Dec-Extreme', 'Dec-Extreme-FullNETs', 'Dec-Extreme-NoNNE'; see Table 1 and Fig. 1).

**Table 1 Description of our scenarios and comparison with the IPCC SR1.5.**

| | Scenario | Comparable primary energy scenario in IPCC SR1.5 | Key characteristics regarding RE and NETs | Trajectory of global GDP 2020–2040 |
|---|---|---|---|---|
| Low decoupling | Degrowth | LED | Strong RE growth, low NETs, no CCS | Shrinks ~−0.5% p.a. |
| | Degrowth-FullNETs | SSP1-1.9/LED | Moderate RE growth, high NETs, no CCS | Shrinks ~−0.2% p.a. |
| | Degrowth-NoNNE | LED | Utopian RE growth, very low NETs, no CCS | Shrinks ~−0.5% p.a. |
| | DLE | None | Below moderate RE growth, very low NETs, no CCS | Shrinks ~−4% p.a. |
| Medium decoupling | Moderate | SSP5-1.9 | Moderate RE growth, high NETs | Grows, ~3.5% p.a. |
| | Moderate-FullNETs | SSP5-1.9 | Below moderate RE growth, very high NETs | Grows, ~3.5% p.a. |
| | Strong | SSP5-1.9 | Strong RE growth, medium NETs | Grows, ~3.5% p.a. |
| | Extreme | SSP5-1.9 | Extreme RE growth, medium NETs | Grows, ~3.5% p.a. |
| | Utopian | SSP5-1.9 | Utopian RE growth, low NETs | Grows, ~3.5% p.a. |
| | IPCC | IPCC SR1.5 median/SSP2-1.9 | Utopian RE growth, low NETs | Grows, ~2.3% p.a. |
| | IPCC-FullNETs | IPCC SR1.5 median/SSP2-1.9 | Moderate RE growth, high NETs | Grows, ~2.3% p.a. |
| | IPCC-NoNNE | IPCC SR1.5 median/SSP2-1.9 | Above utopian RE growth, very low NETs | Grows, ~2.3% p.a. |
| | ClimateAnalytics | None/SSP2-1.9 | Utopian RE growth, medium NETs | Grows, ~2.3% p.a. |
| High decoupling | Dec-Moderate | SSP2-1.9 | Moderate RE growth, high NETs | Grows, ~2.3% p.a. |
| | Dec-Strong | LED | Strong RE growth, low NETs | Grows, ~2.4% p.a. |
| | Dec-Extreme | LED | Extreme RE growth, low NETs | Grows, ~2.4% p.a. |
| | Dec-Extreme-FullNETs | LED | Moderate RE growth, high NETs | Grows, ~2.4% p.a. |
| | Dec-Extreme-NoNNE | LED | Above utopian RE growth, very low NETs | Grows, ~2.4% p.a. |

For the low energy-GDP decoupling group, GDP growth rates (market exchange rate, MER, constant 2010 US$) result from our modelled final energy pathways, combined with average historical (1969–2019) energy-GDP decoupling. Historical GDP data are taken from the World Bank. For the scenarios in the other two groups, GDP growth rates are assumed to equal the rates of the comparable IPCC SR1.5 archetypes linked above in Table 1. Here, we take GDP growth rates in purchasing power parity (PPP, 2010 US$) from the IAMC 1.5 °C Scenario Explorer hosted by IIASA and transform them, following Brockway et al.[8], into MER growth rates using a conversion factor of 0.78, in order to match our historical GDP growth rates in MER. *Dec* decoupling, *RE* renewable energy, *CCS* carbon capture and storage applied to coal and gas, *SSP* shared socioeconomic pathway, *DLE*: decent living energy.

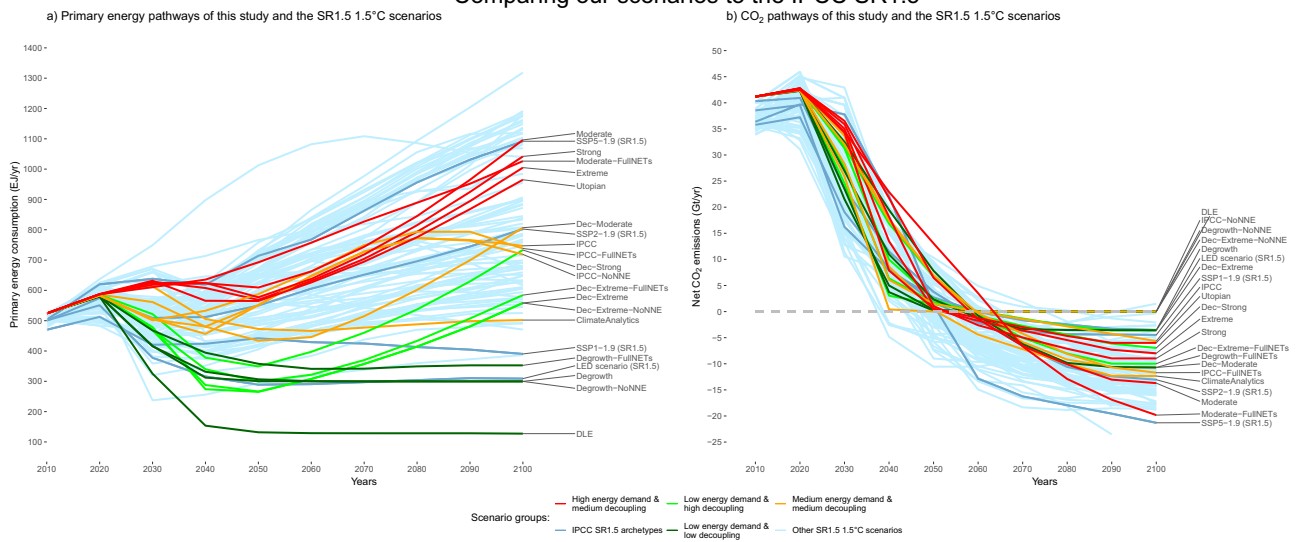

**Fig. 1 Comparison of scenarios in this study to the IPCC SR1.5.** Comparison of the primary energy **a** and net CO$_2$ **b** pathways of the IPCC SR1.5 1.5 °C scenarios and our scenarios. Data for IPCC pathways are taken from the IAMC 1.5 °C Scenario Explorer hosted by IIASA.

In the first group, GDP follows approximately the final energy demand curve (low relative energy-GDP decoupling; see Table 1). In the latter two groups, it is assumed that GDP continues to grow at current growth rates (between 2.3 and 3.5% p.a. as in the SSPs), with slower growth (relative energy-GDP decoupling) or falling final energy demand (absolute energy-GDP decoupling). 'NoNNE' stands for 'no net negative emissions', where only residual emissions from cement and flaring are removed by

NETs, whereas 'FullNETs' stands for high NETs deployment and slower renewable energy growth. 'Dec' stands for higher energy-GDP decoupling. The respective primary and final energy demands mirror selected archetypes from IAM scenarios in the IPCC SR1.5 (see Fig. 1 and Table 1 for a juxtaposition), which mostly show energy-GDP decoupling[2]. For instance, our 'IPCC' scenarios closely follow the median primary energy demand trajectory of the IPCC SR1.5 scenarios, whereas our 'Degrowth' and 'Dec-Extreme' scenarios follow the energy trajectory of the

**Table 2 Summary of the main results of the different scenarios.**

| | Scenario | Net $CO_2$ 2018–2100 (GtCO$_2$) | Max. OS (GtCO$_2$) | RE scale increase 2050/2019 | CDR: start date, max. rate (GtCO$_2$/a), date of max. rate, cumulative CDR (GtCO$_2$) | Cumulative CCS (GtCO$_2$) |
|---|---|---|---|---|---|---|
| Low decoupling | Degrowth | 580 | 134 | 25-fold | 2051, −3.6, 2071, 143 | 0 |
| | Degrowth-FullNETs | 580 | 336 | 18-fold | 2041, −11, 2082, 432 | 0 |
| | Degrowth-NoNNE | 580 | 0 | 27-fold | 2041, −0.45, 2043, 9 | 0 |
| | DLE | 577 | −3 | 11-fold | 2041, −0.91, 2044, 32 | 0 |
| Medium decoupling | Moderate | 579 | 385 | 43-fold | 2041, −14, 2093, 481 | 58 |
| | Moderate-FullNETs | 580 | 470 | 10-fold | 2030, −24, 2074, 1186 | 164 |
| | Strong | 580 | 286 | 52-fold | 2046, −9, 2089, 299 | 50 |
| | Extreme | 579 | 226 | 53-fold | 2051, −8, 2094, 229 | 46 |
| | Utopian | 579 | 180 | 53-fold | 2053, −6, 2088, 183 | 40 |
| | IPCC | 579 | 116 | 52-fold | 2060, −6, >2100, 117 | 27 |
| | IPCC-FullNETs | 580 | 305 | 40-fold | 2044, −12, 2094, 384 | 68 |
| | IPCC-NoNNE | 579 | 0 | 52-fold | 2039, −0.29, 2040, 4 | 0 |
| | ClimateAnalytics | 351 | 185 | 41-fold | 2041, −13, 2089, 473 | 0 |
| | Dec-Moderate | 576 | 322 | 26-fold | 2042, −11, 2085, 412 | 48 |
| High decoupling | Dec-Strong | 580 | 179 | 30-fold | 2054, −7, 2095, 186 | 26 |
| | Dec-Extreme | 577 | 94 | 24-fold | 2061, −3.6, 2087, 98 | 17 |
| | Dec-Extreme-FullNETs | 578 | 291 | 13-fold | 2044, −10, 2088, 346 | 59 |
| | Dec-Extreme-NoNNE | 580 | 0 | 24-fold | 2060, −0.06, 2061, 1 | 0 |

*Dec* decoupling, *OS* overshoot of the carbon budget, *RE* renewable energy, *CDR* cumulative carbon dioxide removal until 2100, *CCS* carbon capture and storage applied to coal and gas, *DLE* decent living energy.

LED scenario[3]. Among the reported IPCC IAM scenarios, the LED scenario is the closest analogue to a degrowth pathway, as it features a strong reduction in material and energy use[18]. The crucial difference here is that our 'Degrowth' scenarios do not rely on technological efficiency measures leading to substantial energy- and material-use-GDP decoupling. At last, the scenarios differ in the speed and scale of renewable energy replacing fossil fuels (from levels below 'Moderate' to above 'Utopian') as well as carbon capture and storage (CCS) and NETs deployment (see Table 1 and Methods).

**Scenario results**. In this simplified representation, all scenarios are designed to stay within the carbon budget for a 50% probability of limiting global temperature rise to below 1.5 °C by 2100 (580 GtCO2[2]). However, Table 2 and Figs. 1–4 show that they achieve this under substantially different primary and final energy consumption, GDP and $CO_2$ emission pathways. Pathways not included in these figures can be found in Supplementary Figs. 1–3. In Fig. 5, we position all our scenarios as well as the IPCC SR1.5 LED, SSP1, SSP2 and SSP5 archetypes on a scenario map along three dimensions: the degree of energy-GDP decoupling, the speed of renewable energy expansion (the 2020–2040 annual average growth in solar, wind and other renewables in EJ/yr) and the level of cumulative NETs and CCS deployment. In Supplementary Fig. 4 (see also Supplementary Tables 1–4) we show a conceptually equivalent figure for a carbon budget of 1170 GtCO$_2$ (>66% chance for 2 °C in 2100[2]) to make the analysis also broadly applicable to reaching the 2 °C target. In the following, we shortly state the main results, following the three groups of energy-GDP decoupling.

First, the low energy-GDP decoupling group (<1.45%, Fig. 5) is the only group showing rates of energy-GDP decoupling that lie within the range of historically experienced values of the rolling 10-year averages of the past 30 years. Regarding the speed of renewable energy expansion there is a wider range, depending on the scale of NETs deployment: with 432 GtCO$_2$ NETs renewables increase at 1.1 EJ/yr (18-fold by 2050, 'Degrowth-FullNETs'), whereas without any net negative emissions they increase at 3.7

EJ/yr (27-fold by 2050, 'Degrowth-NoNNE'). However, the 'DLE' scenario with very low energy demand manages without any net negative emissions, whilst being closest to historical data (0.9 EJ/yr, 11-fold by 2050 and 32 GtCO$_2$ NETs).

The medium energy-GDP decoupling group (1.45–3%, Fig. 5) shows higher longer-term energy-GDP decoupling than historically experienced as per Fig. 5, as has also been found by others[8,24,27]. For the same level of renewable energy increase, the medium-energy-demand group shows lower NETs deployment than the high energy-demand group (e.g., 'IPCC': 144 GtCO$_2$ vs. 'Utopian': 222 GtCO$_2$). It generally holds that the more NETs are deployed the slower renewables are allowed to expand to reach 1.5 °C ('Moderate-FullNETs': 1350 GtCO$_2$, 10-fold and 1.5 EJ/yr vs. 'IPCC-NoNNE': 4 GtCO$_2$, 52-fold and 6.7 EJ/yr).

The high energy-GDP decoupling group (>3%, Fig. 5) shows that the higher the energy-GDP decoupling and the lower the final energy consumption, the slower renewables need to expand and the fewer NETs need to be deployed to reach 1.5 °C ('Moderate': 539 GtCO$_2$, 43-fold and 3.1 EJ/yr vs. 'Dec-Extreme-FullNETs': 346 GtCO$_2$, 13-fold and 0.4 EJ/yr). Next, we assess what these results imply with respect to relative risks for feasibility and sustainability.

**Scenario assessment: interpretation of relative risk indicators**. In choosing our indicators, we broadly follow Loftus et al.[24], e.g., in assessing energy intensity (here energy-GDP decoupling) and the magnitude of renewable energy additions (here relative increase 2019–2050 and absolute average growth rate 2020–2040). We choose absolute indicators instead of relative ones (e.g., normalising absolute renewable energy growth by GDP), because the latter hide important aspects of socio-technical feasibility (e.g., the magnitude of coordination, material extraction, land use and infrastructure expansion). We further include NETs deployment and equity and qualitatively assess socio-political feasibility, broadly following Jewell & Cherp[25]. From this perspective, we conceptualise Fig. 5 as a relative risk map indicating higher risks for socio-technical feasibility and sustainability with increasing energy-GDP decoupling, speed and scale of fossil

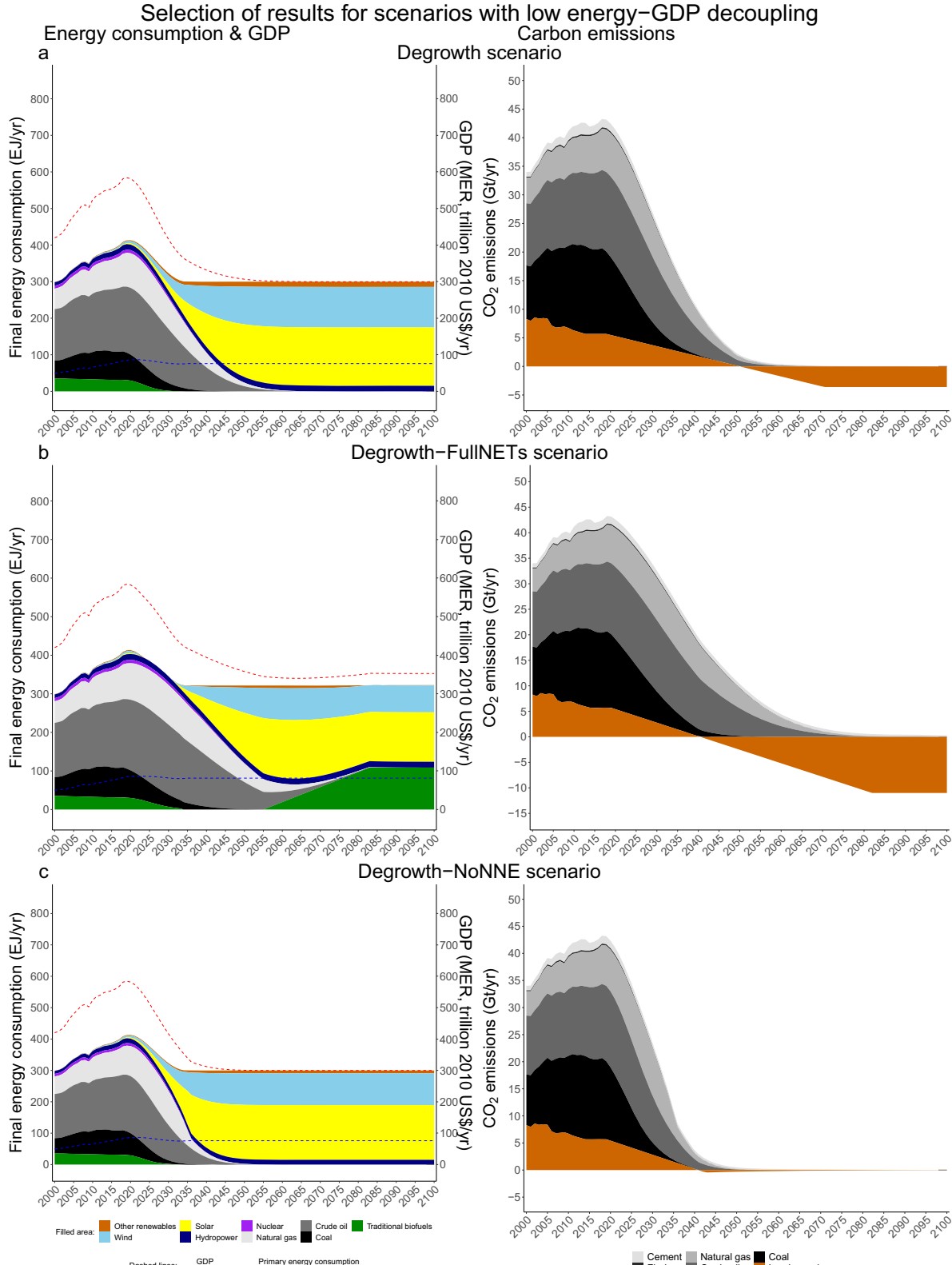

**Fig. 2 Selection of 1.5°C scenarios with low energy-GDP decoupling.** Selection of 1.5 °C scenarios with low energy-GDP decoupling (**a–c**). On the left, final energy consumption (in EJ, left axis), aggregate primary energy consumption (in EJ, red dashed line, left axis) and GDP (MER, in trillion 2010 US$, blue dashed line, right axis). On the right, carbon emissions (in GtCO₂/yr). The full collection of scenarios can be found in Supplementary Fig. 1.

fuel replacement by renewables as well as NETs and CCS deployment. In the following, we review and discuss literature on the importance and interpretation of each risk indicator and assess how our scenarios perform.

**Energy-GDP decoupling**. In the LED scenario, technological efficiency measures such as widespread digitalisation and electrification lead to a 53% reduction in final energy demand in the global North and 32% in the global South (40% globally) by 2050.

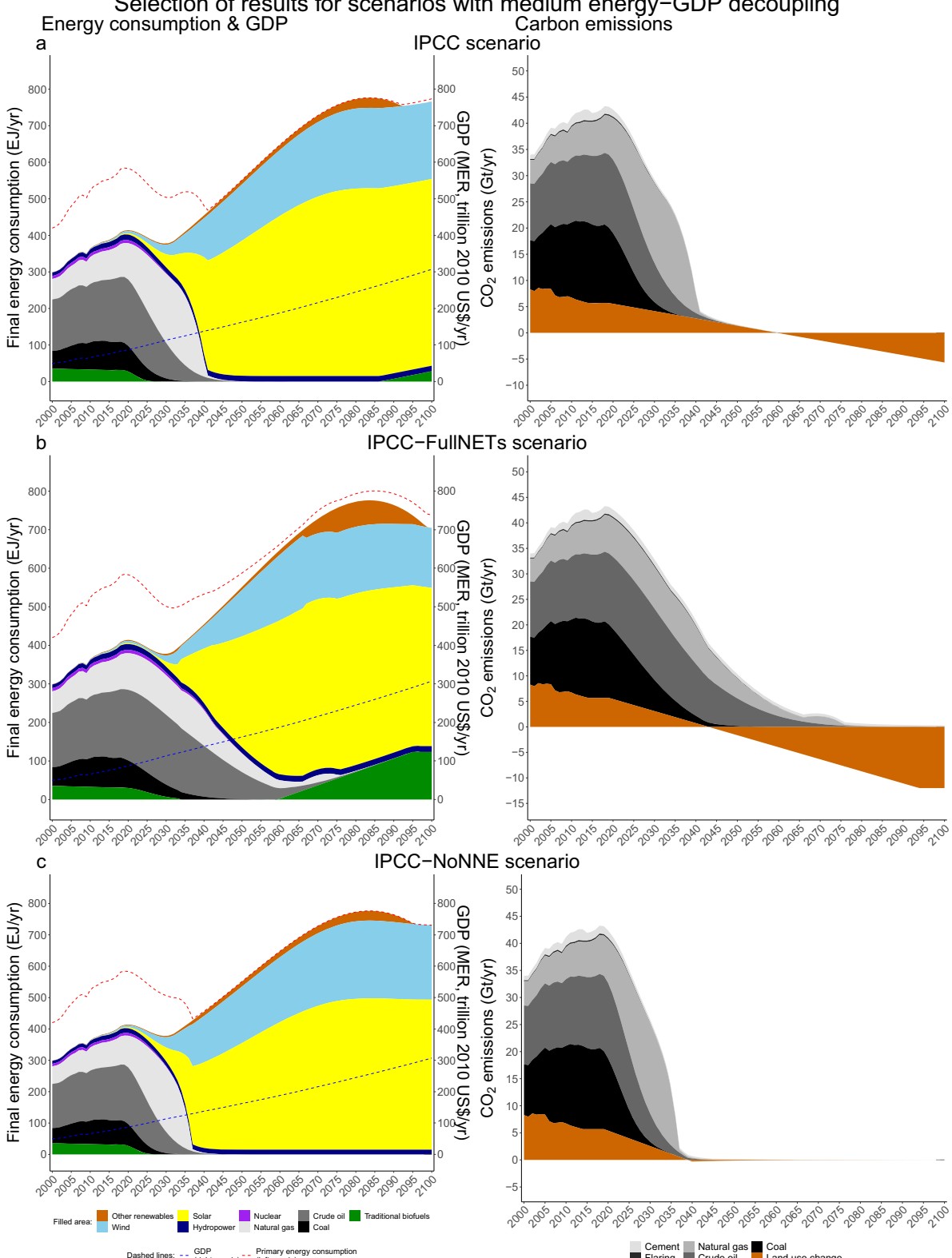

**Fig. 3 Selection of 1.5°C scenarios with medium energy-GDP decoupling.** Selection of 1.5 °C scenarios with medium energy-GDP decoupling (**a–c**). On the left, final energy consumption (in EJ, left axis), aggregate primary energy consumption (in EJ, red dashed line, left axis) and GDP (MER, in trillion 2010 US$, blue dashed line, right axis). On the right, carbon emissions (in GtCO$_2$/yr). The full collection of scenarios can be found in Supplementary Fig. 2.

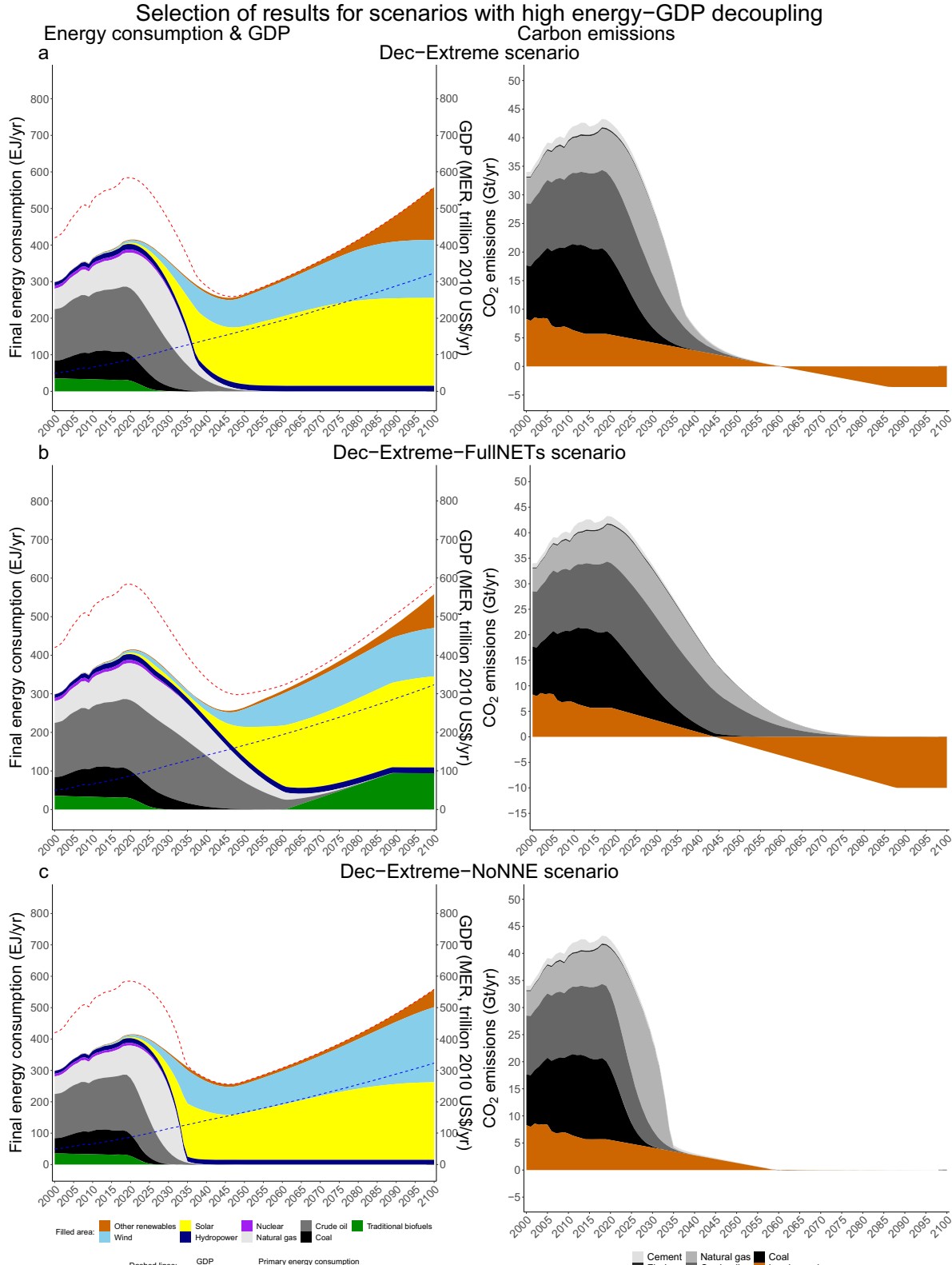

**Fig. 4 Selection of 1.5°C scenarios with high energy-GDP decoupling.** Selection of 1.5 °C scenarios with high energy-GDP decoupling (**a**–**c**). On the left, final energy consumption (in EJ, left axis), aggregate primary energy consumption (in EJ, red dashed line, left axis) and GDP (MER, in trillion 2010 US$, blue dashed line, right axis). On the right, carbon emissions (in GtCO2./yr). The full collection of scenarios can be found in Supplementary Fig. 3.

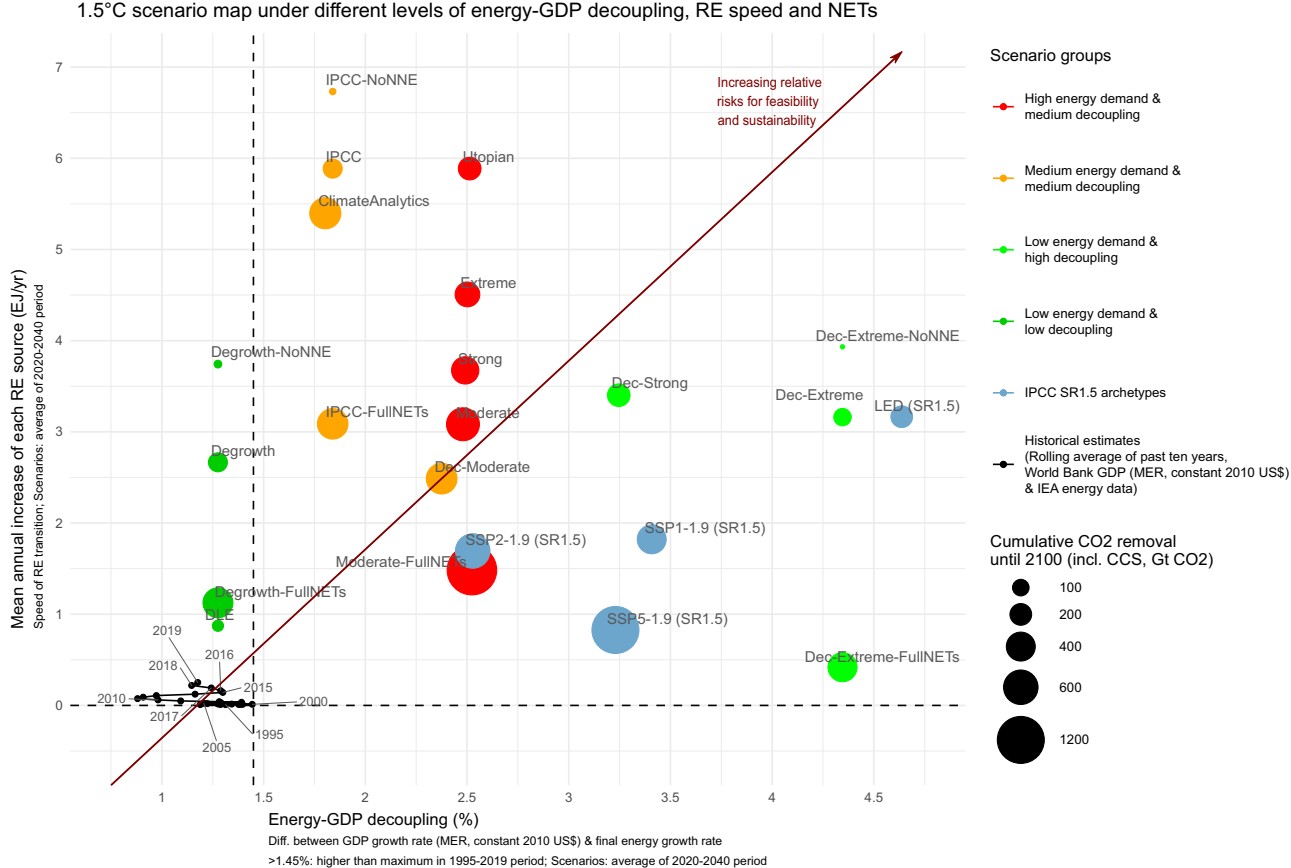

**Fig. 5 1.5 °C scenario map under different levels of energy-GDP decoupling, RE speed and NETs.** The dimensions are 'speed of renewable energy transition' (for the scenarios the 2020–2040 annual average growth in solar, wind and other renewables, in EJ/yr), 'energy-GDP decoupling' (for the scenarios the 2020–2040 average difference between GDP growth rate and final energy growth rate, in %) and cumulative $CO_2$ removal until 2100, including CCS (GtCO$_2$). Historical data points are the rolling averages of the past ten years (e.g., for the 1995 point the period 1986–1995) of the respective indicators. This averaging was chosen (1) because GDP and final energy data are noisy and (2) to emphasise longer-term trends. While historically four years were above a decoupling of 2% since 1986, these are outlieres around a lower, almost constant trend[8]. Historical GDP data (MER, constant 2010 US $) is taken from the World Bank. The conceptually equivalent graph for 2 °C can be found in Supplementary Fig. 4.

However, Grubler et al.[3,5] state that they do not explicitly consider the effect of their energy pathway on GDP growth. In contrast, Hickel[18] calls this scenario a degrowth scenario, owing to the likelihood that such energy-GDP decoupling is impossible. There are several reasons to justify this likelihood, as recently summarised by a number of studies[5,8,14,15,27–29], which are not considered by the IPCC IAMs[8,27]. Firstly, Ayres & Warr[30], Keen et al.[28] and others[7,14,27] show that 'total factor productivity' (other production factors influencing economic growth besides capital and labour) is strongly connected to total energy use and its conversion efficiency into useful energy (energy use after accounting for production and conversion losses), contrary to neoclassical economic theory. Secondly, Sakai et al.[14], find that for industrialised countries such as the UK (p. 1) 'gains in thermodynamic efficiency are a key 'engine of economic growth'' due to economy-wide rebound mechanisms. Thus, (p. 11) '[t]he tight coupling between global energy use and GDP [...] can be explained because of—not in spite of—decades of global energy efficiency investment.' This is in line with recent results by Heun & Brockway[31], who state (p. 1): 'Absolute decoupling of energy from [GDP] appears mission impossible', again owing to feedbacks of efficiency gains. As a recent review[8] concludes, such economy-wide rebound effects undermine more than half of the potential energy efficiency savings. It is further corroborated by recent evidence showing that until now digitalisation has likely

led to a net increase in energy consumption by driving energy efficiency and thus economic growth[19,29]. At last, Ward et al.[15] show that in the longer term (p. 1) 'GDP ultimately cannot plausibly be decoupled from growth in material and energy use'. This is also the case in service-based economies, since services embody materials and energy[29,32] and energy intensive goods are usually outsourced[31]. Increasing tertiarization in industrialised countries has not led to decreases, but rather increases in energy use and $CO_2$ emissions[32]. Biophysical efficiency and scale of an economy appear to be structurally connected[5,6,8,14,27,29,31]. These reasons justify considering the reliance upon high energy-GDP decoupling a substantial risk for feasibility.

From this perspective, the scenarios with the lowest risk for feasibility are those in the 'low energy-GDP decoupling' group, which comprises our 'Degrowth' scenarios. All other scenarios show, in part substantially, higher energy-GDP decoupling than historically experienced as per Fig. 5 (e.g., the LED and our Dec-Extreme scenarios are over three times higher on average between 2020 and 2040).

**Speed and scale of renewable energy replacing fossil fuels.** Firstly, all else unchanged, the higher the necessary speed of increasing renewable energy, the higher is the feasibility challenge[5–7,33]. Second, considering that energy use is strongly coupled to GDP growth, an important measure for the

performance of the energy-economic system is the energy return on energy invested (EROI)[34]. The energy system's EROI is likely to shrink substantially during the transition to a renewable energy system[34] and to remain lower than the EROI of current fossil energy systems afterwards[35]. This is likely to have a limiting effect on GDP growth[23,34]. However, there is a wide range of reported EROI values for individual renewable energy technologies, varying with geographic location and applied methodologies[35]. Moreover, in contrast to fossil fuels, the EROI of renewables is projected to increase over time[36], but there also are counteracting effects, among others by diminishing returns to EROI at higher grid penetration[27,34,35]. Such energy constraints for economic activity are not taken into account by the IAMs reviewed by the IPCC[27]. However, a recent IAM modelling study using the MEDEAS IAM framework finds that such energy constraints are likely to reduce GDP growth[23]. In addition, Floyd et al.[37] summarise 10 points implying deep uncertainties in renewable energy's ability to meet a high and rising energy demand, stating that the lower the energy demand, the higher the likelihood to meet it. Thus, this review points towards a 100% renewable energy economy likely being a smaller one, in GDP and final energy terms.

Third, there is no empirical evidence for the possibility of an absolute decoupling between GDP and aggregate material use[5–7]. The reduction of the latter is central for climate mitigation[38], the reduction of environmental impacts[5] and prevention of biodiversity loss[39]. Large-scale renewable energy deployment is unlikely to contribute to material use reduction[5–7,34], as renewables have a considerably higher material footprint than fossil fuels[6,34]. This may also raise critical risks of metal supply shortages[34]. Further, material extraction drives conflicts with local communities around the world, especially in the global South[34,40]. In order to be more sustainable, the global material footprint would need to be significantly scaled down, to ~50 billion tonnes per year (recognising the limits of aggregate indicators[5]), which is highly unlikely to be compatible with growing GDP[5–7,39]. These three points justify considering the reliance upon high speed and scale of the renewable energy transition a substantial risk for feasibility and sustainability.

As shown by our results, the transition speed depends heavily on the accepted NETs deployment as well as the energy demand. This can well be seen from our three 'FullNETs-X-NoNNE' scenario combinations in Fig. 5. Similarly, regarding the scale, the 'FullNETs' scenarios show the lowest levels. If less NETs are accepted, the scale follows the energy demand level: the two 'Degrowth' scenarios (25-fold and 27-fold) perform similarly to the 'Dec-Extreme' scenarios (24-fold and 24-fold), followed by the 'Dec-Strong' scenario and after that by the medium and high energy demand scenarios.

**Negative emission technologies**. Large-scale NETs deployment faces numerous and substantial risks for sustainability and feasibility[2]. Only two NETs, AR and soil carbon sequestration, are currently available at scale[13]. However, IAMs most prominently include BECCS[12]. In doing so, modellers make numerous assumptions of substantial uncertainty[11,41]. The EROI of BECCS may be extremely low[27]. BECCS is associated with major land-use change and its potentially negative side-effects[2,10,12], e.g., the further transgression of several planetary boundaries[42], especially biodiversity[43]. CCS, either as part of BECCS, or applied to coal and gas, faces similar barriers and uncertainties[2,44]. More risks of reliance on large-scale NETs remain[2,10,12], e.g., direct air capture technologies strongly increasing energy and water use[45]. Even large-scale AR as a NET is not unproblematic, being vulnerable to carbon loss and having potentially negative side-effects on land use change, albedo, biodiversity, and food security[12,13,41].

Anderson & Peters[46] thus conclude that (p. 183) 'the mitigation agenda should proceed on the premise that [NETs] will not work at scale. The implications of failing to do otherwise are a moral hazard par excellence.' Therefore, it is justified to consider the reliance upon large-scale (e.g., medium (200–400 GtCO$_2$) and high (>400 GtCO$_2$)) NETs deployment a substantial risk for feasibility and sustainability.

The scenarios minimising NETs (<200 GtCO$_2$) either show very high renewable growth and medium energy-GDP decoupling ('IPCC' and 'IPCC-NoNNE'), low energy-GDP decoupling and high renewable growth ('Degrowth' and 'Degrowth-NoNNE') or high energy-GDP decoupling and high renewable growth ('Dec-Extreme' and 'Dec-Extreme-NoNNE'). Compared with these scenarios, degrowth scenarios are relying on the lowest speed and scale of renewable energy expansion as well as the lowest energy-GDP decoupling for any shared level of NETs deployment, thus showing the lowest risks for feasibility and sustainability.

**Equity**. Equity is vital for sustainability, as increasing the income of the poorest population segments to above 2.97$ is projected to use 66% of the carbon budget for 2 °C[47], whereas only a global, affluent minority is currently and historically responsible for most carbon emissions[19,48]. This implies later (earlier) peak dates and lower (higher) mitigation rates for low- (high-) income countries[33]. When including equity, as stated throughout the Paris Agreement, the short-term mitigation agenda for high-income countries becomes substantially more challenging than is deemed feasible by IAMs, assuming continued GDP growth, thus implying the need for degrowth in the global North[5,33]. Generally, all our scenarios do not consider the global distribution of energy consumption. However, taking into account the above environmental justice perspective is especially important for the equitable downscaling of throughput in the 'Degrowth' scenarios[9]. Thus, and to obtain a first impression of potential distributional consequences, we present a scenario for the energy use distribution between global South and North for two of our scenarios in Fig. 6. Here, we assume an equal per capita distribution of global energy use in 2050 among 10 billion people, as is modelled by Millward-Hopkins et al.[20] to be approximately (±≈15–20%) the case with respect to global variations in energy use for basic human needs satisfaction. However, aggregate growth scenarios, as in the case of the 'Moderate' scenario, are subject to the limitations and risks of the renewable energy transition, material extraction, the wider ecological crisis and NETs discussed above, thus again neglecting equity[34,40,46]. Therefore, taking equity into account further justifies reconsidering the reliance on high NETs deployment as well as the speed and scale of the renewable energy transition, as the risks for feasibility and sustainability increase. This lends further support to the finding above that 'Degrowth' pathways minimise risks for feasibility and sustainability.

**Political and economic feasibility**. Compared with technology-driven pathways, it is clear that a degrowth transition faces tremendous political barriers[9,49]. As Kallis et al.[9] state, currently (p. 18) '[a]bandoning economic growth seems politically impossible', as it implies significant changes to current capitalist socioeconomic systems in order to overcome its growth imperatives[9,19,49]. Degrowth, moreover, challenges deeply embedded cultures, values, mind-sets[21] and power structures[9,19]. However, as Jewell & Cherp state, political feasibility is softer than socio-technical feasibility[25], with high actor motivation potentially compensating for low action capacity and social change being complex, non-linear and essentially unpredictable[50]. Political feasibility further depends to a large extent on social movements formulating and pushing for the

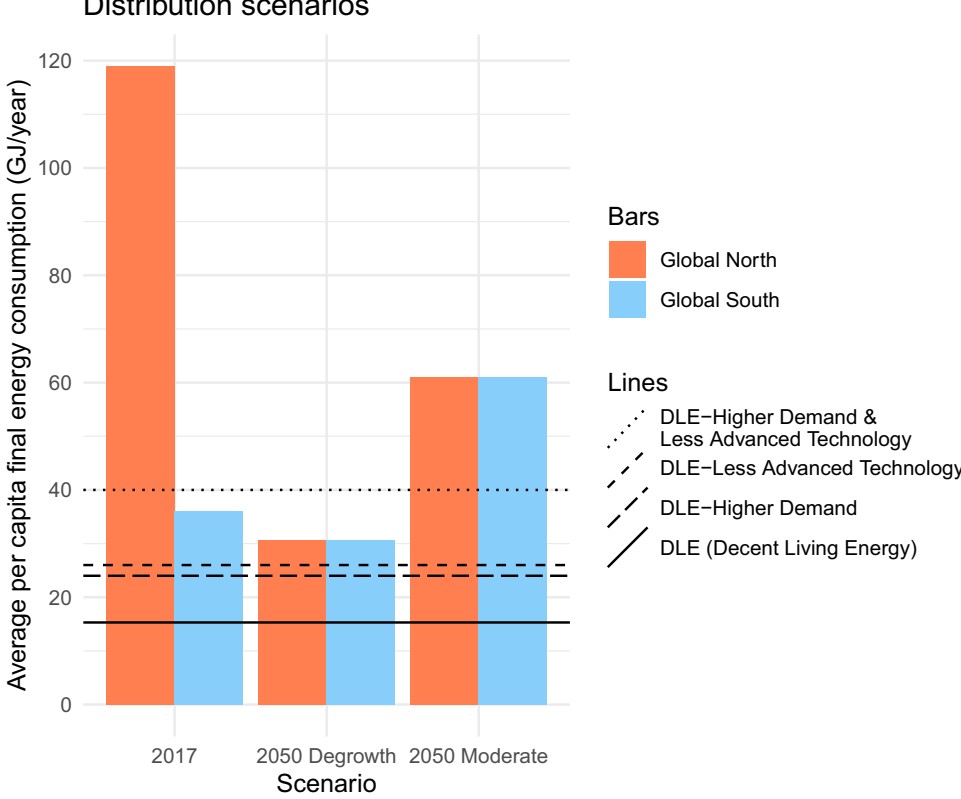

**Fig. 6 A final energy distribution scenario for our 1.5 °C 'degrowth' and 'moderate' scenario, assuming an equal per capita distribution among 10 billion people in 2050.** We additionally include historical data for 2017 and all Decent Living Energy (DLE) scenarios modelled by Millward-Hopkins et al.[20], which give an approximation of the energy use for basic human needs satisfaction under varying assumptions. The global North here comprises the OECD, non-OECD Europe and Eurasia, whereas the South comprises all other regions from the IEA[63]. Note that we use territorial and not consumption-based data, which is likely to understate the differences in 2017[48].

implementation of political programs, changing values, practices and cultures and building alternative institutions[49,51] as well as scientists pointing the way to alternative paradigms[49]. Consequently, degrowth implies modifications to the strategies for change, with a stronger focus on bottom–up social movements[9,19,49]. As many research questions on degrowth remain open[9,19] and the state of political feasibility can change with better knowledge about and awareness of alternative paradigms, strengthened social movements and a clearer understanding about transition processes[49–51], it is even more crucial to investigate degrowth pathways.

'Economic feasibility' usually refers to the monetary costs of a mitigation pathway, reported as share of GDP[4,24]. Here, many IAMs follow a cost-minimisation approach in order to maximise economic welfare[4,41], measured in GDP, by progressively implementing only the mitigation measures with the lowest marginal abatement costs. From this perspective, degrowth is often considered economically inefficient, as the GDP loss is considered a cost and when weighted with the avoided $CO_2$ appears to be overly expensive compared to technological measures[52]. However, this reasoning presupposes a fictitious 'optimal' GDP growth path, any negative deviation from which is a priori defined as a 'cost'[41]. Importantly, GDP is not a neutral construct[9]. Thus, one needs to ask to whom costs occur, who profits, whose contributions are in- or excluded and finally who should decide this[9]. So, even if this GDP loss is accepted as a 'cost', this reasoning compares two categories that have very different, and partly incommensurable, welfare implications. For instance, the costs of replacing a coal plant with wind turbines

(a technological measure: creating jobs and reducing $CO_2$, but using land and materials) are not directly monetarily comparable to the costs of producing, consuming as well as working less (a GDP loss: polluting less, while using less resources and potentially leading to further positive social consequences, if well managed). To have a more valid comparison between the two categories, one would need to monetise the whole variety of ecological and social impacts on different groups of people and ecosystems, which is impossible at least without strong value judgements[4,9]. A more suitable perspective when dealing with climate justice issues in a wellbeing context is a human needs provisioning approach[16,20,53]. The crucial question then becomes how, if GDP were to shrink as a result of the required reductions in material and energy use and $CO_2$ (the degrowth hypothesis), e.g., through stringent eco-taxes and/or caps[9], this GDP decrease could be made socially sustainable, i.e. safeguarding human needs and social function[9,21]. Here, research shows that in principle it is possible to achieve a high quality of life with substantially lower energy use and GDP[9,16,17,20]. As noted in the introduction, however, substantial socioeconomic changes would be necessary to avoid the effects of a recession. Moreover, the reductions and limits would need to be democratically negotiated[9,21,49] and consider potential 'sufficiency rebound effects'[54] (reduced consumption by some being compensated through increases by others), e.g., by international coordination.

To summarise, as indicated by Fig. 5, the 1.5 °C degrowth scenarios have the lowest relative risk levels for socio-technical feasibility and sustainability, as they are the only scenarios relying in combination on low energy-GDP decoupling, comparably low

speed and scale of renewable energy replacing fossil fuels as well as comparably low NETs and CCS deployment. When excluding any NETs and CCS deployment, the degrowth scenarios still show the lowest levels of energy-GDP decoupling as well as speed and scale of renewable energy replacing fossil fuels, compared to the 'IPCC' and 'Dec-Extreme' pathways. As a drawback, degrowth scenarios currently have comparably low socio-political feasibility and require radical social change. This conclusion holds as well for the 2 °C scenarios, albeit with less extreme differences. Here, the 'Degrowth-NoNNE' scenario, with ~0% p.a. global GDP growth, is almost aligned with historical data, in stark contrast to the technology-driven scenarios without net negative emissions (see Supplementary Fig. 4).

## Discussion

The results indicate that degrowth pathways exhibit the lowest relative risks for feasibility and sustainability when compared with established IPCC SR1.5 pathways using our socio-technical risk indicators. In comparison, the higher the technological reliance of the assessed mitigation pathways, the higher the risks for socio-technical feasibility and sustainability. The reverse is likely the case for socio-political feasibility, which, however, is softer than socio-technical feasibility. This result contrasts strongly with the absolute primacy of technology-driven IAM scenarios in the IPCC SR1.5. In what follows, we discuss limitations of our modelling approach and risk indicators, implications for the IAM community and further research.

Our results face several limitations. Note that we use the carbon budget for a 50% chance to stay below 1.5 °C[2], which can be argued to be too low based on the precautionary principle, especially when considering that such scenarios still include a 10% chance of reaching catastrophic warming of 3 °C[55]. Already increasing the chance for 1.5 °C to 66% lowers the available carbon budget by 160 $GtCO_2$, while including earth system feedbacks lowers it by an additional ~100 $GtCO_2$[2]. In addition, note that we do not consider $CH_4$ and $N_2O$ emissions, for which technological mitigation is more problematic than for $CO_2$. Including all these factors would substantially increase the mitigation challenges. Any such increase further strengthens the case for considering degrowth scenarios, since it becomes even more risky to solely rely on technology to accomplish the higher mitigation rates. Thus, complementing technology by far-reaching demand reductions through social change becomes even more necessary for 1.5 °C to remain feasible. This is especially the case when considering the softer nature of social feasibility compared with socio-technical feasibility. We nevertheless stress that feasibility is a highly complex concept that can be interpreted differently and, in the case of individual scenarios, remains at least in part subjective[24]. Therefore, a larger variety of indicators than ours is certainly necessary to arrive at a more complete picture of feasibility. However, we maintain that such research should explicitly consider degrowth scenarios, e.g., along the lines of the 'Societal Transformation Scenario' by Kuhnhenn et al.[56] or the 'SSP0' scenario proposed by Otero et al.[39]. Especially in view of socio-political feasibility, we argue that not exploring them actually leads to a self-fulfilling prophecy: with research subjectively judging such scenarios as infeasible from the start, they remain marginalised in public discourse, thus inhibiting social change, thus letting them appear as even more infeasible to the scientist and so on. As McCollum et al.[57] and Pye et al.[58] argue, modellers have a collective responsibility to evaluate the full spectrum of future possibilities, including scenarios commonly deemed politically unlikely.

A further limitation of this study is our simplified quantitative model, which only addresses the fuel-energy-emissions nexus

top–down. This enhances transparency and understanding and is suited for the purpose of this study by allowing to assess relative feasibility. Moreover, it enables modelling pathways currently excluded by the IPCC IAMs, avoiding the difficulties and complexities with modelling degrowth (see below and Methods). Nevertheless, our model neglects the monetary sector[22], connections between energy and material availability and economic growth[23,34] as well as the bottom–up energy and material requirements for decent living standards[20]. This potentially renders some scenarios infeasible, despite our efforts to qualitatively include these factors in our above treatment of feasibility. Therefore, our simplified modelling approach can only be a very first step to exploring degrowth scenarios and needs to be complemented by more complex modelling.

To our knowledge, no in-depth study examining the reasons for the omission of degrowth scenarios in mainstream IAM modelling exists (but see[4]). Such modelling is highly challenging, partly because a degrowth society would function differently compared to the current society. Thus, model parameters and structures based on past data could no longer be valid[59]. Furthermore, it would need to recognise that GDP is an inadequate indicator for societal wellbeing, at least in affluent countries. Instead, the focus needs to be oriented directly at multi-dimensional human needs satisfaction[9,18,53]. This is especially important given that many degrowth proposals include a strengthening of non-monetary work, such as care work and community engagement, as well as decommodification of economic activity towards sharing, gifting and commons[9,59]. This also implies revisiting the widespread, neoclassical economic optimisation approach in IAMs[4,23,59]. More plural economic perspectives would need to be taken into account to gain a fuller picture of socioeconomic reality[22,59,60], e.g., post-Keynesian, ecological and Marxian economics. Such modelling would also need to broaden the considered portfolio of demand-side measures and behavioural changes[4,61,62]. At last, it is clear that the biophysical foundation of economic activity and energy efficiency rebound effects need to be considered in much greater detail[8,23,27]. The necessary detailed discussion of how exactly IAMs would need to change to incorporate some of these features is beyond the scope of this paper, but such discussions are already under way in the literature[8,27,58,61,62] and could be further inspired by current efforts in ecological macroeconomic modelling[59]. Promising developments in these directions are put forward by the MEDEAS IAM modelling framework, which connects biophysical economic insights, system dynamics and input–output analysis[23,34]. Another recent example is the EUROGREEN model, combining post-Keynesian and ecological economics in a system dynamics stock-flow consistent framework to assess socio-ecological consequences of national degrowth and green growth scenarios[22].

In light of the optimism of IAM mitigation scenarios regarding technological change[3,8,27], NETs[2,46] as well as the neglect of the wider ecological crisis[5–7,39] and equity issues[33,40,46], it should be a priority to explore alternative scenarios. Clearly, degrowth would not be an easy solution, but, as indicated by our results, it would substantially minimise many key risks for feasibility and sustainability compared with established, technology-driven pathways. Therefore, it should be as widely and thoroughly considered and debated as are comparably risky technology-driven pathways.

## Methods

In this work, we project global $CO_2$ as the sum of a number of components: (i) $CO_2$ from fuel combustion, (ii) $CO_2$ from cement manufacturing and flaring and (iii) $CO_2$ from forestry and land use change. Our approach is simplified in that it only addresses the fuel-energy-emissions nexus, and in that it uses heuristic rather than theory-driven parametrisations. However, this simplified approach enhances transparency and understanding, whereas it suffices for the purpose of this article:

to compare degrowth with IPCC SR1.5 scenarios using different risk indicators. With this approach, we react to the summary of criticisms of complex IAMs by Gambhir et al.[26] and follow their proposed 'supplement' strategy, complementing complex IAMs with a simpler, but still 'fit-for-purpose' model. Moreover, with this approach, we are able to model degrowth and other pathways which are not included in the IPCC SR1.5. However, we simultaneously avoid the issues and complexities of modelling degrowth in greater detail, e.g., monetary or biophysical aspects, which are subject to further research[23,59]. We recognise the many limitations of this approach (see Discussion), but maintain that it is suited for the purpose of this study and in calling attention to the need for more research on degrowth pathways. Our study is a first step in this direction. A full version of our model can be accessed in Supplementary Data 1.

**Final and primary energy demand and CO$_2$ emissions from fuel combustion.** Future CO$_2$ emissions from fuel combustion are derived from future global primary energy demand, which is in turn derived from future global final energy demand $e_c(t)$ of $c = 1,...,10$ energy carriers (1: natural gas, 2: coal, 3: crude oil, 4: nuclear energy, 5: traditional biofuels, 6: hydro-electricity, 7: solar PV, 8: wind energy, and 9: other renewable energy (concentrating solar, wave, geothermal etc), 10: total final energy). The final energy demands are converted to primary energy using constant 2017 conversion efficiencies from IEA data[63]. Throughout this paper, we will call fuels 1–5 "conventional" and 6–9 "renewable") for the period $t = 2019,...,2050$. We recursively model future final energy demand as

$$e_c(t) = [1 + \gamma_c(t)]e_c(t-1) \text{ with change rates } \gamma_c(t) = \gamma_c(t-1) + \delta_c \quad (1)$$

Here, $\gamma_c(t)$ are carrier- and time-dependent annual rates of change, and $\delta_c$ are carrier-dependent annual change rate increments. Given that carrier 10 is the sum over individual carriers $c = 1,...,9$, the annual energy balance of this recursive scheme is

$$\sum_{c=1}^{9} e_c(t) = e_{10}(t). \quad (2)$$

With recursively determined $\gamma_c$ and $\delta_c$, Eq. 2 does in general not hold. To ensure the balance holds, we adjust the demand of certain fuels at each annual iteration, giving rise to two cases: (1) If the sum of individual carrier demands $\sum_{c=1}^{9} e_c(t)$ exceeds the prescribed total $e_{10}(t)$, we downscale the combined demand $\sum_{c=7}^{9} e_c(t)$ for solar, wind and other renewables so that it equals the residual of total final energy demand minus conventional fuels $e_{10}(t) - \sum_{c=6}^{6} e_c(t)$. Condition (1) can occur towards the end of the projection period where the rapid growth of renewables cannot be absorbed by demand, and their output needs to be curtailed. (2) If the prescribed total $e_{10}(t)$ exceeds the sum of individual carrier demands $\sum_{c=1}^{9} e_c(t)$, we upscale the demand $e_1(t)$ for natural gas so that it picks up the slack $e_{10}(t) - \sum_{c=2}^{9} e_c(t)$. Condition (2) can occur towards the beginning of the projection period where the rapid decommissioning of coal and oil outpaces the growth of renewables, leaving gaps in supply that need to be filled with natural gas. The recursive scheme in Eq. 1 is seeded with demand values $e_c(t = 2019)$ and rates of change $\gamma_c(t = 2019)$.

**CO$_2$ emissions from fuel combustion.** CO$_2$ emissions from fuel combustion $f_c(t)$ were derived from primary energy demand through fuel-specific emissions coefficients $\varphi_c$:

$$f_c(t) = e_c(t)\varphi_c \quad (3)$$

**CO$_2$ process emissions.** There is a range of emissions sources that are unrelated to energy demand. We model CO$_2$ emissions $g(t)$ from cement manufacturing and flaring through constant change rates $\beta$ as

$$g(t) = g(t = 2019)(1 + \beta)^{t-2019}, \quad (4)$$

as improved technology for processes is assumed to be able to mitigate an increasing amount of emissions. Finally, we model CO$_2$ emissions $l(t)$ from forestry and land use through constant annual increments $\lambda$ as

$$l(t) = \max[-24\text{Gt}, l(t = 2019) + \lambda(t - 2019)], \quad (5)$$

which reflects constant ongoing efforts of re- and afforestation and the possibility for negative emissions.

**Data sources.** Historical records of primary energy demand $e_c(t)$ by energy carrier were taken from the IEA[64] and used for deriving seed rates of change $\gamma_c(t = 2019)$. Historical CO$_2$ emissions $f(t)$ from fuel combustion were taken from CDIAC[65]. Historical emissions $g(t)$ and $l(t)$ for CO$_2$ from cement manufacturing, flaring, land use and forestry are from FAOSTAT[66].

**Scenarios.** The scenarios are designed as archetypes, broadly covering the range of primary energy demand, renewable energy growth, energy-GDP decoupling and negative emissions of the IPCC SR1.5 (see Fig. 1, Fig. 5 and Table 1 for a comparison). This is done to ensure that our results are broadly applicable to established IAM scenarios reviewed in the IPCC SR1.5.

We investigate four pathways with low energy-GDP decoupling (the consumption-driven degrowth pathways: 'Degrowth', 'Degrowth-FullNETs', 'Degrowth-NoNNE' and 'DLE'), 10 scenarios with medium energy-GDP decoupling (the technology-driven scenarios: 'Moderate', 'Moderate-FullNETs', 'Strong', 'Extreme', 'Utopian', 'IPCC', 'IPCC-FullNETs', 'IPCC-NoNNE', 'ClimateAnalytics' and 'Dec-Moderate'), as well as four technology-driven pathways with high energy-GDP decoupling (called 'Dec-Strong', 'Dec-Extreme', 'Dec-Extreme-FullNETs', 'Dec-Extreme-NoNNE'; see Table 1 and Fig. 1). In the technology-driven pathways 'Moderate', 'Moderate-FullNETs', 'Strong', 'Extreme' and 'Utopian', total final energy demand is seeded with a 2019 growth rate of $\gamma_c(t = 2019) = 1\%$ (based on the New Policies Scenario of the IEA[67]: The 2017 IEA World Energy Outlook states that "in the New Policies Scenario, global energy needs rise more slowly than in the past but still expand by 30% between today and 2040." This trajectory is equivalent to a constant ($\delta_c = 0$) annual growth of 1% over 23 years. In order to increase the variety of energy pathways, we further annually increase this growth rate by 0.004% ('Moderate'), 0.003% ('Strong'), 0.002% ('Extreme') and 0.001% ('Utopian'). In all the later pathways (including 'Moderate-FullNETs') we assume a GDP growth rate at constant energy (MER, constant 2010 US$) equal to the one of SSP5-1.9. Here, we take GDP growth rates in purchasing power parity (PPP, 2010 US$) from the IAMC 1.5 °C Scenario Explorer hosted by IIASA and transform them, following Brockway et al.[8], into MER growth rates using a conversion factor of 0.78, in order to match our historical GDP growth rates in MER. All respective scenario parameters can be found in Tables 1, 3–6.

In the two technology-driven pathways 'Moderate' and 'Dec-Moderate', coal initially declines at a rate of 2% per year, and this rate is declining further at –0.3% per year. Similarly, by decreasing their rates of change by –0.3% per year, oil and gas turn from moderate growth in 2018 to a decline by 2028. Change rates for traditional biofuels are ramped down as well at –1% per year. We keep hydro-electricity constant, because its global potential is deemed to be exhausted considering competing uses, anticipated increases in drought frequency, and concerns surrounding ecological health[68–70]. As of 2015 and 2016, mature renewable technologies were growing rapidly at an average annual 29% (solar PV) and 16% (wind), which we use as start values in 2019. To avoid sudden demand gaps, we decreased their rates of growth by 0.75% and 0.3% per year, respectively. Moreover, this is in line with current knowledge about the logistic growth patterns of renewables[71] as well as with increasing issues concerning intermittency, EROI and resources connected to increasing scale[34,37]. According to the International Energy Agency[72], other renewable technologies such as "concentrating solar power, geothermal and ocean technologies are currently not on track with their SDS targets". We, therefore, increase their growth rate by 0.2% per year. The reduction of emissions from cement manufacturing and flaring is technology-driven and occurs at –2% per year. Reductions of annual emissions from forestry and land-use change occur at constant increment of –269 Mt and −254 Mt CO$_2$, respectively, until a cap of –3.6 GtCO$_2$/yr is reached (the maximum potential for AR as a NET according to the IPCC[2]). Thereafter, negative emissions occur in the form of BECCS until the total maximum for these scenarios is reached (–14 and –11 GtCO$_2$/yr, respectively). CCS applied to gas and coal starts in 2029 at 3.33% and increases from 2031 linearly at 1.33% until a maximum of 30% of coal and natural gas usage is reached (within the approximate range for 2050 in the IPCC SR1.5[2]).

The '(Dec-)Strong', '(Dec-)Extreme', 'Utopian' and 'IPCC' technology-driven pathways work similarly, but with coal, oil, gas, nuclear and traditional biofuels being phased out successively more rapidly, growth rates for solar PV and wind being brought down less quickly, or even stabilised until 2050, and—optimistically—other renewables being introduced more rapidly. In addition, industrial emissions are brought down more rapidly. Caps on NETs as well as the speed of emission reductions from forestry and land use differ in order to minimise overshoot in 2100 or during the whole 21st century (the 'NoNNE' scenarios). Generally, AR is scaled up first, followed by BECCS. The latter technology produces primary energy at 18.75 EJ/GtCO$_2$xyr, which is within the range reported by Smith et al.[12] and Fajardy & Dowell[73]. The maximum CCS usage as well as its increase rate increases in accordance with the technological level of the scenario. In the scenarios without net negative emissions, other parameters were adjusted as well to eliminate overshoot during the 21st century. The 'FullNETs' scenarios were run with the 'Moderate' levels of renewable energy replacing fossil fuels, whereas renewable energy expands even slower in the 'Moderate-FullNETs' scenario.

The consumption-driven degrowth pathways with low energy-GDP decoupling are radically different from the technology-driven pathways with medium and high energy-GDP decoupling (see Table 1). Their main feature is that final energy demand turns from an initial growth at 1% per year to an immediate stabilisation the year after, and then into a slow decline. This is achieved by less consumption and production, thus global GDP following the final energy pathway using the average level of decoupling between 1995 and 2019. In order to model a 'societal soft landing', we treat the annual change rate increment as time-dependent,

$$\delta_{FE}(t) = \min(0\%, \delta_{FE}(t = 2019) + \varepsilon(t - 2020)),$$

with

$$\varepsilon = -0.9\% + 0.11\%(t - 2020). \quad (6)$$

This means that initially, the final energy growth rate changes from +1% in 2019 to +0.1% in 2020, to −0.69% in 2021 etc, but then the decline slows down

**Table 3 Parameters for pathways with low energy-GDP decoupling.**

**Low energy-GDP decoupling**

| | Annual change rate increments $\delta_c$ (%) | | | | Seed value (%) |
|---|---|---|---|---|---|
| Carrier | Degrowth | Degrowth-FullNETs | Degrowth-NoNNE | DLE | $\gamma_c(t = 2019)$ |
| Coal | −0.6 | −0.3 | −1.0 | −0.9 | −2.0 |
| Crude oil | −0.6 | −0.3 | −1.0 | −0.9 | 1.8 |
| Natural gas | −0.6 | −0.3 | −1.0 | −0.9 | 2.8 |
| Nuclear | −1.0 | −0.5 | −1.0 | −0.5 | 1.5 |
| Traditional biofuels | −2.0 | −1.0 | −2.0 | −1.0 | −0.5 |
| Hydro-electricity | 0.0 | 0.0 | 0.0 | 0.0 | 0.0 |
| Solar PV | −0.65 | −0.75 | −0.35 | −1.0 | 28.8 |
| Wind | −0.25 | −0.35 | 0.05 | −0.4 | 15.9 |
| Other renewables | 0.3 | 0.2 | 0.6 | 0.2 | 0.5 |
| Total FE demand | −0.9 | −0.9 | −0.9 | −1.3 | 1.0 |
| Constant change rates $\beta$ (%) | | | | | |
| Cement & flaring | −5.0 | −2.0 | −5.0 | −2.0 | |
| Constant annual increments $\lambda$ (Mt CO$_2$/a) | | | | | |
| Forestry & land use | −179 | −265 | −260 | −265 | |
| Maximum rate of negative emissions (GtCO$_2$/a) | | | | | |
| Forestry & land use | −3.6 | −11.0 | 0.0 | 0.0 | |
| Maximum CCS share of energy from coal and natural gas (%) | | | | | |
| Coal & gas | 0.0 | 0.0 | 0.0 | 0.0 | 0.0 |
| Linear increase rate of CCS (%/a) | | | | | |
| Coal & gas | 0.0 | 0.0 | 0.0 | 0.0 | 0.0 |

PV photovoltaic, FE final energy. Carrier-dependent annual change rate increments $\delta_c$ and seed values $\gamma_c(t = 2019)$ for annual rates of change as in Eq. 1; constant change rates $\beta$ as in Eq 4; constant annual increments $\lambda$ as in Eq. 5.

**Table 4 Parameters for pathways with medium energy-GDP decoupling (1).**

**Medium energy-GDP decoupling (1)**

| | Annual change rate increments $\delta_c$ (%) | | | | | Seed value (%) |
|---|---|---|---|---|---|---|
| Carrier | Moderate | Moderate-FullNETs | Strong | Extreme | Utopian | $\gamma_c(t = 2019)$ |
| Coal | −0.3 | −0.1 | −0.6 | −0.9 | −1.2 | −2.0 |
| Crude oil | −0.3 | −0.1 | −0.6 | −0.9 | −1.2 | 1.8 |
| Natural gas | −0.3 | −0.1 | −0.6 | −0.9 | −1.2 | 2.8 |
| Nuclear | −0.5 | −0.5 | −1.0 | −1.5 | −2.0 | 1.5 |
| Traditional biofuels | −1.0 | −1.0 | −2.0 | −3.0 | −5.0 | −0.5 |
| Hydro-electricity | 0.0 | 0.0 | 0.0 | 0.0 | 0.0 | 0.0 |
| Solar PV | −0.75 | −0.75 | −0.65 | −0.55 | −0.4 | 28.8 |
| Wind | −0.3 | −0.35 | −0.25 | −0.15 | −0.05 | 15.9 |
| Other renewables | 0.2 | 0.2 | 0.3 | 0.4 | 0.5 | 0.5 |
| Total FE demand | 0.004 | 0.0 | 0.003 | 0.002 | 0.001 | 1.0 |
| Constant change rates $\beta$ (%) | | | | | | |
| Cement & flaring | −2.0 | −2.0 | −5.0 | −7.5 | −7.5 | |
| Constant annual increments $\lambda$ (Mt CO$_2$/a) | | | | | | |
| Forestry & land use | −269 | −551 | −211 | −183 | −170 | |
| Maximum rate of negative emissions in GtCO$_2$/a | | | | | | |
| Forestry & land use | −14.0 | −24.0 | −9.0 | −8.0 | −6.0 | |
| Maximum CCS share of energy from coal and natural gas (%) | | | | | | |
| Coal & gas | 30.0 | 30.0 | 35.0 | 40.0 | 45.0 | 0.0 |
| Linear increase rate of CCS (%/a) | | | | | | |
| Coal & gas | 1.33 | 1.33 | 1.58 | 1.83 | 2.08 | 0.0 |

PV photovoltaic, FE final energy, CCS carbon capture and storage. Carrier-dependent annual change rate increments $\delta_c$ and seed values $\gamma_c(t = 2019)$ for annual rates of change as in Eq. 1; constant change rates $\beta$ as in Eq 4; constant annual increments $\lambda$ as in Eq. 5.

until it stops in 2035 from where final energy use is constant. In our results we show how this change profile harmonises with the replacement of fossil fuels by renewables (Fig. 2). In the degrowth pathways, fossil phase-outs and non-combustion emission reductions are assumed to range between moderate ('Degrowth-FullNETs'), strong ('Degrowth') and between extreme and utopian ('Degrowth-NoNNE').

At last, the technology-driven pathways with high and medium energy-GDP decoupling, which reduce final energy consumption work similarly to the degrowth scenarios. The key difference to the degrowth scenarios is that here it is assumed that GDP continues to grow at current growth rates, equal to the GDP (MER) growth rates of the LED scenario or SSP2-1.9 respectively (see Table 1), despite falling or stagnating final energy demand (strong relative or absolute decoupling),

**Table 5 Parameters for pathways with medium energy-GDP decoupling (2).**

**Medium energy-GDP decoupling (2)**

| Carrier | Annual change rate increments $\delta_c$ (%) | | | | | Seed value (%) |
| | IPCC | IPCC-FullNETs | IPCC-NoNNE | Climate Analytics | Dec-Moderate | $\gamma_c(t = 2019)$ |
|---|---|---|---|---|---|---|
| Coal | −1.2 | −0.3 | −1.7 | −1.2 | −0.3 | −2.0 |
| Crude oil | −1.2 | −0.3 | −1.7 | −1.2 | −0.3 | 1.8 |
| Natural gas | −1.2 | −0.3 | −1.7 | −1.2 | −0.3 | 2.8 |
| Nuclear | −2.0 | −0.5 | −2.0 | −1.0 | −0.5 | 1.5 |
| Traditional biofuels | −5.0 | −1.0 | −5.0 | −2.0 | −1.0 | −0.5 |
| Hydro-electricity | 0.0 | 0.0 | 0.0 | 0.0 | 0.0 | 0.0 |
| Solar PV | −0.40 | −0.73 | 0.00 | −0.40 | −0.75 | 28.8 |
| Wind | −0.05 | −0.34 | 0.35 | −0.05 | −0.30 | 15.9 |
| Other renewables | 0.5 | 0.2 | 0.9 | 0.6 | 0.2 | 0.5 |
| Total FE demand | −0.7 | −0.7 | −0.7 | −0.05 | −0.15 | 1.0 |
| Constant change rates $\beta$ (%) | | | | | | |
|   Cement & flaring | −7.5 | −2.0 | −7.5 | −1.0 | −2.0 | |
| Constant annual increments $\lambda$ (Mt $CO_2$/a) | | | | | | |
|   Forestry & land use | −140 | −236 | −285 | −270 | −254 | |
| Maximum rate of negative emissions in Gt$CO_2$/a | | | | | | |
|   Forestry & land use | −6.0 | −12.0 | 0.0 | −13.0 | −11.0 | |
| Maximum CCS share of energy from coal and natural gas (%) | | | | | | |
|   Coal & gas | 45.0 | 45.0 | 0.0 | 0.0 | 30.0 | 0.0 |
| Linear increase rate of CCS (%/a) | | | | | | |
|   Coal & gas | 2.08 | 2.08 | 0.0 | 0.0 | 1.33 | 0.0 |

PV photovoltaic, FE final energy, CCS carbon capture and storage. Carrier-dependent annual change rate increments $\delta_c$ and seed values $\gamma_c(t = 2019)$ for annual rates of change as in Eq. 1; constant change rates $\beta$ as in Eq 4; constant annual increments $\lambda$ as in Eq. 5.

**Table 6 Parameters for pathways with high energy-GDP decoupling.**

**High energy-GDP decoupling**

| Carrier | Annual change rate increments $\delta_c$ (%) | | | | Seed value (%) |
| | Dec-Strong | Dec-Extreme | Dec-Extreme-FullNETs | Dec-Extreme-NoNNE | $\gamma_c(t = 2019)$ |
|---|---|---|---|---|---|
| Coal | −0.6 | −0.9 | −0.3 | −2.4 | −2.0 |
| Crude oil | −0.6 | −0.9 | −0.3 | −2.4 | 1.8 |
| Natural gas | −0.6 | −0.9 | −0.3 | −2.4 | 2.8 |
| Nuclear | −1.0 | −1.5 | −1.5 | −2.0 | 1.5 |
| Traditional biofuels | −2.0 | −3.0 | −3.0 | −5.0 | −0.5 |
| Hydro-electricity | 0.0 | 0.0 | 0.0 | 0.0 | 0.0 |
| Solar PV | −0.65 | −0.55 | −0.75 | 0.25 | 28.8 |
| Wind | −0.25 | −0.15 | −0.35 | 0.65 | 15.9 |
| Other renewables | 0.3 | 0.4 | 0.2 | 1.1 | 0.5 |
| Total PE demand | −0.3 | −0.5 | −0.5 | −0.5 | 1.0 |
| Constant change rates $\beta$ (%) | | | | | |
|   Cement & flaring | −5.0 | −7.5 | −7.5 | −7.5 | |
| Constant annual increments $\lambda$ (Mt $CO_2$/a) | | | | | |
|   Forestry & land use | −167 | −138 | −228 | −141 | |
| Maximum rate of negative emissions in Gt$CO_2$/a | | | | | |
|   Forestry & land use | −7.0 | −3.6 | −10.0 | 0.0 | |
| Maximum CCS share of energy from coal and natural gas (%) | | | | | |
|   Coal & gas | 35.0 | 40.0 | 40.0 | 0.0 | 0.0 |
| Linear increase rate of CCS (%/a) | | | | | |
|   Coal & gas | 1.58 | 1.83 | 1.83 | 0.0 | 0.0 |

PV photovoltaic, FE final energy, CCS carbon capture and storage. Carrier-dependent annual change rate increments $\delta_c$ and seed values $\gamma_c(t = 2019)$ for annual rates of change as in Eq. 1; constant change rates $\beta$ as in Eq 4; constant annual increments $\lambda$ as in Eq. 5.

achieved by technological efficiency improvements. We model the 'societal soft landing' similarly to Eq. 6, but instead of 0, −0.9 and 0.11% we use different parameters: 1.5, −0.15 and 0.008% ('Dec-Moderate'), 1.5, −0.3 and 0.02% ('Dec-Strong') and 1.5, −0.7/−0.5 and 0.035% ('Dec-Extreme', 'Dec-Extreme-FullNETs' and 'Dec-Extreme-NoNNE'). Moreover, the 'IPCC' scenario group as well as the 'ClimateAnalytics' scenario are special cases where the factors are empirical

parameters, with, in the former case, Eq. 6 becoming

$$\delta_{FE}(t) = \min\left(2.2\% - 4.5\varepsilon(t - 2020)^2,\right.$$
$$\left.\min\left(2.5\%, \delta_{FE}(t = 2019) + \varepsilon(t - 2020)\right)\right),$$

with

$$\varepsilon = -0.7\% + 0.12\% \, (t - 2020). \qquad (7)$$

in order to make it as similar as possible to the IPCC SR1.5 median primary energy pathway. In the latter case, Eq. 6 becomes

$$\delta_{\mathrm{FE}}(t) = \max\left(0, \min\left(1.5\%, \delta_{\mathrm{FE}}(t = 2019) + \varepsilon(t - 2020)\right)\right),$$

with

$$\varepsilon = -0.05\% + 0.001\% \, (t - 2020). \qquad (8)$$

**Reporting summary**. Further information on research design is available in the Nature Research Reporting Summary linked to this article.

## Data availability

All relevant data underpinning our modelling are cited throughout the study and Methods. We further deposit a full version of our model, as described in the Methods, in Supplementary Data 1.

## Code availability

The R code used to create the figures is available upon request from the corresponding author.

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

## Acknowledgements

This work was supported by the Australian Research Council through its Discovery Projects DP0985522 and DP130101293, and Linkage Infrastructure, Equipment and Facilities Grant LE160100066, and the National eResearch Collaboration Tools and Resources project (NeCTAR) through its Industrial Ecology Virtual Laboratory VL201. Sebastian Juraszek expertly managed our advanced computation requirements, and Charlotte Jarabak from SciTec Library collected data. Lastly, we gratefully acknowledge comments on this manuscript by Samuel Alexander (University of Melbourne), Stefan Pauliuk (University of Freiburg), Ursula Fuentes (Climate Analytics), Viktoria Cologna and Giulia Fontana (ETH Zurich) as well as two anonymous reviewers. Remaining errors are our own.

## Author contributions

L.K. crafted the qualitative discussion, M.L. developed the quantitative model, M.L and L.K. designed the scenarios, analysed the data and wrote the paper.

## Competing interests

The authors declare no competing interests.
