## [Peer Review File · Nature Communications]

REVIEWER COMMENTS

Reviewer #1 (Remarks to the Author):

This is an original and important paper. The authors begin from the observation that existing IPCC climate mitigation scenarios all entail continued GDP growth at a global level. This introduces a problem, because GDP growth entails rising energy demand, which makes it difficult to achieve a sufficiently rapid transition to renewable energy to remain within carbon budgets for 1.5C or 2C. As a result, the vast majority of scenarios have no choice but to rely on high levels of net negative emissions, which are increasingly considered to be dangerous and unfeasible at the scale that these scenarios require. The authors address this problem by building a "degrowth" scenario, where global energy demand is scaled down, making it easier to achieve full renewable energy coverage and zero emissions by 2050.

Up to now, the only scenario that has approximated this approach is the LED scenario by Grubler et al (2018), in IPCC SR1.5. But the LED scenario has a number of problems, specifically, it relies on high levels of efficiency improvements without modeling rebound effects. Existing empirical evidence suggests that the efficiency improvements that Grubler et al assume are likely to lead to more energy and material use, not less, so long as the economy remains organized around GDP growth. In other words, Grubler et al get part of the answer right (namely, that IPCC scenarios need to significantly reduce energy and material use in order to be consistent with 1.5 or 2C), but they do not adequately explain how this could be accomplished. The "degrowth" model answers the latter question. This paper helps to solve a longstanding problem, and I am thrilled to see that this work has been done. It represents a significant contribution, and I believe it will influence thinking in the field.

The paper is well-grounded in the literature on degrowth.

I do however have a few suggestions for improvement.

First, the maximum global energy reduction you consider is to 300 EJ/year, which is roughly a 50% reduction from existing levels. I see that there is a recent paper in Global Environmental Change that indicates a 60% reduction can be achieved while ending global poverty and providing a good standard of living for all at a population of 10 billion in 2050. <https://www.sciencedirect.com/science/article/pii/S0959378020307512?via=ihub> It would be worth citing this paper, and perhaps even modeling it. This additional reduction of energy demand would take further pressure off rapid renewable outlay, thus further improving technological feasibility.

Second, I think it would be worth specifying, as Grubler et al do, how these demand reductions would be reasonably divided up between global North and South. Clearly some poor countries will need to increase energy demand, while the vast majority of reduction will have to occur in the global North. I believe that the GEC paper linked above may help to answer this question.

Third, it might be addressing the question of 2C budgets, even if only briefly. I can imagine some might assume that shifting to a 2C budget removes the need for degrowth. Is that the case? Or do your conclusions apply to 2C as well?

Fourth, it seems to me that your paper could be framed more effectively. Right now you open your abstract by saying you will seek to solve the problem of the absence of degrowth scenarios used by the IPCC. This is not a good lead, because most people will not consider this to be a problem. Instead, you should lead with the fact that all existing rely heavily on NNEs and technological changes, because they start from the assumption that growth must continue forever. But this is unfeasible. So, that's the problem that needs to be solved. And the alternative degrowth scenario offers a workable solution.

You do a better job with this in the introduction to the text, but still I think it could be framed more effectively. You write:

"Meanwhile, integrated assessment model (IAM) mitigation scenarios reported by the Intergovernmental Panel on Climate Change (IPCC) Special Report on 1.5°C (SR1.5) rely on controversial amounts of carbon dioxide removal and/or on unprecedented technological changes^{2,3}. Alternative mitigation pathways as examined by the expanding degrowth literature⁴ are almost completely neglected by the IAM community and the IPCC^{2,5}."

 again, it is not the fact that degrowth literature is neglected by the IPCC that is the problem. It is the existing scenarios that are the problem. Use this opening paragraph to describe why.

 why do the scenarios rely on so much CO₂ removal and tech change? Because they assume perpetual growth.

 why do they assume perpetual growth? Because they assume that growth is necessary for the economy to function and deliver well-being.

I think you should lead by explaining this in a few sentences, or in a paragraph. Then you can say that a degrowth scenario would solve this problem, and note that thus far the IPCC has neglected degrowth scenarios. Your intervention is therefore to redress this.

Finally, probably you should say a bit more about living standards at the beginning of the paper. Again, one of the key reasons that IAMs rely on perpetual growth is because this is considered necessary for ending poverty and improving livelihoods etc. So you should counter this assumption earlier on (right now you get to it at line 397, I believe). Briefly say *how* it is possible to have a reduction of global GDP of 0.3% per year while still ending poverty and providing a decent life for all (i.e., how this will not lead to mass impoverishment).

Reviewer #2 (Remarks to the Author):

The paper includes various degrowth scenarios as a way for the world to meet the 1.5-degree goal, as outlined in the Paris agreement. By including degrowth scenarios, excluded in the IPCC reports, the authors claim to make a novel contribution. They construct their scenarios along three dimensions: degree of energy-GDP decoupling, the speed of renewable energy expansion, and the cumulative NETs and CCS deployment. Although the authors make a conceptual difference between "feasibility", "sustainability", and "political feasibility", they should have defined their concept in the introduction, especially the difference between "feasibility" and "political feasibility". As a point of departure in these concepts, they make the argument that current IPCC scenarios are too optimistic with regards to technological fixes. They also argue that the trust in CCS (carbon capture and storage) and NETs (net emissions technologies) and the possibility for decoupling energy use and GHG emissions from continuous economic growth (measured as GDP) are too optimistic. Technological fixes are not proven on a large scale and the large uncertainties inherent with CCS and NETs, so other mitigation pathways such as degrowth should be given attention.

Their point of departure is the importance of including degrowth scenarios since "not exploring them actually leads to a self-fulfilling prophecy: with research subjectively judging such scenarios as 'infeasible' from the start, they remain marginalized in public discourse". The authors put forward well-known criticism from an ecological economics point of view towards models based on a more orthodox and neoclassical point of view, especially with regards to the possibilities for decoupling energy use from GDP. The authors could have made their epistemological point of view more evident and made it evident that they criticize the more orthodox and neoclassical economic models' dependence on economic growth and GDP.

They point on the limitations in the service-based economy from embodied energy and material use as well as outsourcing connected to transferring production of goods to other countries. They further claim that large-scale renewable energy deployment is unlikely to contribute to material use reduction, since renewables have a considerably higher material footprint than fossil fuels. Their critique centers around the following: 1) Current modelers consider degrowth to be subjectively 'implausible' because GDP is viewed as a suitable indicator of human well-being, it has negative social consequences, and the assumption that it also applies to the global South; 2) The

limited representation of behavioral change within integrated assessment modelling compared to the strong supply-side technology focus, thus neglecting demand-side changes that lead to sufficiency; and 3) Some integrated assessment modelling optimizes welfare functions connected to GDP growth and is thus fundamentally unsuited to modelling degrowth.

They propose that the focus should be oriented directly at multidimensional human needs satisfaction. Many degrowth proposals include a strengthening of non-monetary work, such as care work and community engagement, as well as a decommodification of economic activity towards sharing, gifting, and commons. Their proposal also implies a revisiting of the widespread optimization approach in IAMs. The authors claim that this approach implies a plural economic perspective, including along heterodox lines (such as post-Keynesian, ecological, and Marxian economics), to gain a fuller picture of socio-economic reality. Such modelling would also need to broaden the considered portfolio of demand-side measures and behavioral changes to complement the current supply-side technology focus. They also claim that the biophysical foundation of economic activity needs to be considered in much greater detail.

The novelty of the paper is not the arguments per se, which are well-known in other contexts, but in the application of degrowth scenarios to the mitigation of GHG emissions. The paper will be of interest for a broad set of readers interested in pathways for mitigating GHG emissions according to the target set in the Paris agreement. However, the authors do not clarify how to expand the IAMs or the epistemological differences from more orthodox economic thinking. Is it possible to make a radical shift in IAMs and change assumptions, or is it impossible to integrate ecological economic thinking along with neoclassical economics? The authors argue that the expansion in current models must include wider perspectives but do not discuss the model and disciplinary opportunity and barriers for such an expansion.

Treatment of the degrowth scenarios and of the degrowth concept itself was difficult to grasp. Do they mean a sustainable degrowth, implying not recession but planned downscaling, which also considers intra- and intergenerational equity? The authors also do not mention that degrowth implies different transfer and rebound effects. For example, sufficiency rebound was addressed by Alcott (2010). Individuals and nations who live with less consumption and production create rebound effects that drive consumption and production elsewhere, thus calling for a global degrowth policy and strategy, which also could involve a fairer distribution between countries. Alcott (2010) also argued that the only way to curb rebound effects was to place absolute physical caps on natural resources, such as keeping oil and gas in the ground; if society first caps its resources, people will automatically live more efficiently and sufficiently (Alcott 2014). The authors could have made it more evident what a degrowth policy would mean at a global level, and what transfer effects and unintended effects from such a policy would look like. Here, the author could have expanded their discussion. In that context also reflections from the current COVID-19 situation, on the connection between GDP and GHG-emissions and on how society could be rebuilt post Covid-19 could have been added to their discussion see for example Le Quéré et. al (2020).

The paper is written clearly and in good English. All key assumptions of the paper and how their scenarios are constructed are made evident for the reader and in a way that is possible for others to replicate.

I suggest that the paper could be published in a revised form if the authors make it more evident whether IAMs can expand along their suggested lines, add better definitions of their concepts in the introduction, and provide clearer and broader explanations of degrowth in their context, also including rebound effects and unintended effects from such a policy.

Literature

- Alcott, B., 2010. Impact caps: why population, affluence and technology strategies should be abandoned. *Journal of Cleaner Production* 18(6), 552-560.
- Alcott (2014) Rebound effects In D'Alisa, G., Demaria, F., & Kallis, G. (Eds.). (2014). *Degrowth: a vocabulary for a new era*. Routledge.
- Sorrell, S.; Dimitropoulos, J. *Ukerc Review of Evidence for the Rebound Effect: Technical Report 5—Energy Productivity and Economic Growth Studies*; UK Energy Research Centre: London, UK,

2007. (This report makes an in-depth discussion on the difference between ecological economics and neo-classical/orthodox economy on models and drivers for economic growth).

Le Quéré, Corinne, et al. "Temporary reduction in daily global CO₂ emissions during the COVID-19 forced confinement." *Nature Climate Change* (2020): 1-7.

Response to Reviewers' Comments

> Our responses are preceded by the > sign as well as formatted in blue colour. We also provide a track-changed version of the revised manuscript additionally to a clean one. All our references to lines in the manuscript refer to the clean PDF version.

Reviewer #1 (Remarks to the Author):

This is an original and important paper. The authors begin from the observation that existing IPCC climate mitigation scenarios all entail continued GDP growth at a global level. This introduces a problem, because GDP growth entails rising energy demand, which makes it difficult to achieve a sufficiently rapid transition to renewable energy to remain within carbon budgets for 1.5C or 2C. As a result, the vast majority of scenarios have no choice but to rely on high levels of net negative emissions, which are increasingly considered to be dangerous and unfeasible at the scale that these scenarios require. The authors address this problem by building a "degrowth" scenario, where global energy demand is scaled down, making it easier to achieve full renewable energy coverage and zero emissions by 2050.

Up to now, the only scenario that has approximated this approach is the LED scenario by Grubler et al (2018), in IPCC SR1.5. But the LED scenario has a number of problems, specifically, it relies on high levels of efficiency improvements without modeling rebound effects. Existing empirical evidence suggests that the efficiency improvements that Grubler et al assume are likely to lead to more energy and material use, not less, so long as the economy remains organized around GDP growth. In other words, Grubler et al get part of the answer right (namely, that IPCC scenarios need to significantly reduce energy and material use in order to be consistent with 1.5 or 2C), but they do not adequately explain how this could be accomplished. The "degrowth" model answers the latter question. This paper helps to solve a longstanding problem, and I am thrilled to see that this work has been done. It represents a significant contribution, and I believe it will influence thinking in the field.

The paper is well-grounded in the literature on degrowth.

> We thank you very much for these supportive words and your constructive comments.

I do however have a few suggestions for improvement.

First, the maximum global energy reduction you consider is to 300 EJ/year, which is roughly a 50% reduction from existing levels. I see that there is a recent paper in *Global Environmental Change* that indicates a 60% reduction can be achieved while ending global poverty and providing a good standard of living for all at a population of 10 billion in 2050.

<https://www.sciencedirect.com/science/article/pii/S0959378020307512?via=ihub> It would be worth citing this paper, and perhaps even modeling it. This additional reduction of energy demand would take further pressure off rapid renewable outlay, thus further improving technological feasibility.

> Thank you for raising this point. We followed your request and cited this new paper by Millward-Hopkins et al. (2020) throughout our manuscript (see e.g. line 82, 390 and 462). We further included an additional Degrowth scenario which approximates the global final energy use of their ‘Decent Living Energy’ (DLE) scenario. The CO₂ and energy pathways of this new scenario are shown in Supplementary Figure 1 and it is included in Figure 1 and 5 as well as Tables 1 and 2. Interestingly, our ‘Degrowth’ scenario already approximates the ‘DLE’ scenarios with relaxed assumptions (the ‘DLE-Higher Demand’ and ‘DLE-Less Advanced Technology’ scenarios; see also new Figure 6).

Second, I think it would be worth specifying, as Grubler et al do, how these demand reductions would be reasonably divided up between global North and South. Clearly some poor countries will need to increase energy demand, while the vast majority of reduction will have to occur in the global North. I believe that the GEC paper linked above may help to answer this question.

> Thank you for this comment. We followed your request and added a new distributional scenario on a per capita basis in Figure 6 in the section on ‘Equity’. Figure 6 is preceded by the following new paragraph (lines 383-391):

“Generally, all our scenarios do not consider the global distribution of energy consumption. However, taking into account the above environmental justice perspective is especially important for the equitable downscaling of throughput in the ‘Degrowth’ scenarios⁸. Thus, and to obtain a first impression of potential distributional consequences, we present a scenario for the energy use distribution between global South and North for two of our scenarios in Figure 6. Here, we assume an equal per capita distribution of global energy use in 2050 among 10 billion people, as is modelled by Millward-Hopkins et al.¹⁸ to be approximately ($\pm\approx 15\text{-}20\%$) the case with respect to global variations in energy use for basic human needs satisfaction.”

Within Figure 6, we further compare our per capita distributional scenario to all four scenarios modelled in Millward-Hopkins et al. (2020) to get an impression of how they compare to the energy needed for human needs satisfaction in their scenarios. As can be seen in Figure 6, in the ‘Degrowth’ scenario the global North by far shows the most substantial reductions (~75% less than in 2017), while the global South only reduces by 15%. Even in our aggregate growth scenario ‘Moderate’, the global North reduces by 50% while the South approximately doubles. It is important to note, as we do in the legend for Figure 6, that we only use territorial data for 2017, which likely underestimates the differences due to global trade.

Third, it might be addressing the question of 2C budgets, even if only briefly. I can imagine some might assume that shifting to a 2C budget removes the need for degrowth. Is that the case? Or do your conclusions apply to 2C as well?

> Thank you for pointing this out. We followed your request and repeated our modelling for a 2°C budget of 1170 GtCO₂. The resulting conceptual equivalent to Figure 5 can be found in Supplementary Figure 4, as well as the respective model parameters in Supplementary Tables 1-4. In our main manuscript, we further include the following new text at the beginning of our results section (lines 192-194):

“In Supplementary Figure 4 (see also Supplementary Tables 1-4) we show a conceptually equivalent figure for a carbon budget of 1170 Gt CO₂ (>66% chance for 2°C in 2100²) to make the analysis also broadly applicable to reaching the 2°C target.”

Further, we write at the end of the results section (lines 481-484):

“This conclusion holds as well for the 2°C scenarios, albeit with less extreme differences. Here, the ‘Degrowth-NoNNE’ scenario, with 0.1% p.a. global GDP growth, is almost aligned with historical data, in stark contrast to the technology-driven scenarios without net negative emissions (see Supplementary Figure 4).”

From these results it is clear that also for 2°C a degrowth scenario would minimize the assessed risks for feasibility and sustainability. Thus, our conclusion holds here as well. As stated above, globally this would imply an approximately constant GDP as of today (only 0.1% growth p.a.), while for the global North, following Figure 6, this would clearly mean a strong reduction from current levels of energy use and GDP.

Fourth, it seems to me that your paper could be framed more effectively. Right now you open your abstract by saying you will seek to solve the problem of the absence of degrowth scenarios used by the IPCC. This is not a good lead, because most people will not consider this to be a problem. Instead, you should lead with the fact that all existing rely heavily on NNEs and technological changes, because they start from the assumption that growth must continue forever. But this is unfeasible. So, that's the problem that needs to be solved. And the alternative degrowth scenario offers a workable solution.

You do a better job with this in the introduction to the text, but still I think it could be framed more effectively. You write:

"Meanwhile, integrated assessment model (IAM) mitigation scenarios reported by the Intergovernmental Panel on Climate Change (IPCC) Special Report on 1.5°C (SR1.5) rely on controversial amounts of carbon dioxide removal and/or on unprecedented technological changes^{2,3}. Alternative mitigation pathways as examined by the expanding degrowth literature⁴ are almost completely neglected by the IAM community and the IPCC^{2,5}."

 again, it is not the fact that degrowth literature is neglected by the IPCC that is the problem. It is the existing scenarios that are the problem. Use this opening paragraph to describe why.

 why do the scenarios rely on so much CO₂ removal and tech change? Because they assume perpetual growth.

 why do they assume perpetual growth? Because they assume that growth is necessary for the economy to function and deliver well-being.

I think you should lead by explaining this in a few sentences, or in a paragraph. Then you can say that a degrowth scenario would solve this problem, and note that thus far the IPCC has neglected degrowth scenarios. Your intervention is therefore to redress this.

> We thank you for highlighting this. We followed your request and changed the framing in the Abstract as well as in the Introduction. However, we were more careful with raising a direct causality between the models' reliance on economic growth and reliance on negative emissions and technological change. We do this on the one hand because we believe that this causality is more complex, as shortly described by Kuhnenn (Ref. #4), making it difficult to summarize shortly, and on the other hand we believe that doing so would partly already presuppose some of our results, namely that degrowth scenarios have the potential to substantially lower the risks regarding

negative emissions and technological change. The relevant part of the Abstract now reads as follows (lines 13-20):

“1.5°C scenarios reported by the Intergovernmental Panel on Climate Change (IPCC) rely on combinations of controversial negative emissions and unprecedented technological change, but simultaneously assume continued growth in gross domestic product (GDP). In contrast, degrowth scenarios, where GDP shrinks due to stringent climate change mitigation, are mostly neglected. Thus, their potential to avoid reliance on negative emissions and technological change remains unexplored. As a first step to address this gap, this paper compares 1.5°C degrowth scenarios with IPCC archetype scenarios, using a simplified quantitative representation of the fuel-energy-emissions nexus.”

Additionally, the relevant part of the Introduction reads as follows (new text formatted in italic and underlined; lines 35-48):

“Five years after the Paris Agreement, CO₂ emissions are still rising¹, and mitigation timelines for the 1.5°C and 2°C climate target are becoming ever more stringent². Meanwhile, integrated assessment model (IAM) mitigation scenarios reported by the Intergovernmental Panel on Climate Change (IPCC) Special Report on 1.5°C (SR1.5) rely on controversial amounts of carbon dioxide removal and/or on unprecedented technological changes^{2,3}. *Simultaneously, all of them assume continued growth in gross domestic product (GDP), among other reasons because this is deemed necessary to support societal wellbeing⁴. However, continued GDP growth is widely associated with increasing mitigation challenges, e.g. by increasing energy and material consumption⁵⁻⁷.* In contrast, alternative mitigation pathways as examined by the expanding degrowth literature⁸, are almost completely neglected by the IAM community and the IPCC^{2,4}. *Thus, their potential to avoid negative emissions and technological change remains unexplored.* In this paper, we present an in-depth comparison of IPCC IAM and degrowth mitigation scenarios by applying a simplified quantitative model of the fuel-energy-emissions nexus.”

Finally, probably you should say a bit more about living standards at the beginning of the paper. Again, one of the key reasons that IAMs rely on perpetual growth is because this is considered necessary for ending poverty and improving livelihoods etc. So you should counter this assumption earlier on (right now you get to it at line 397, I believe). Briefly say *how* it is possible to have a reduction of global GDP of 0.3% per year while still ending poverty and providing a decent life for all (i.e., how this will not lead to mass impoverishment).

> Thank you for raising this point. We followed your request and briefly specified potential wellbeing implications of degrowth as well as ways to ensure a high quality of life and poverty eradication in the introduction, while deleting the respective discussion later in the paper. The relevant part now reads as follows (lines 76-85):

“On wellbeing, research^{15,16} shows that high-income countries could scale back their biophysical impact (and GDP), while maintaining (or even increasing^{8,17}) social performance and achieving higher equity among countries. Thus, intra- and intergenerational equity aspects can be taken into account^{8,16,18}, e.g. by making the world economy structurally fairer and redistributing from global North to South^{16,17}. Further, bottom-up studies show that high living standards can be maintained with substantially less per capita energy use than currently consumed in affluent countries¹⁹. However, to ensure that such reductions do not lead to the socially harmful and inequitable effects of a recession requires deep socio-economic changes and policy reforms, such as universal basic services, maximum incomes, working time reductions and democratic firm ownership^{8,16,18,20}.”

Reviewer #2 (Remarks to the Author):

The paper includes various degrowth scenarios as a way for the world to meet the 1.5-degree goal, as outlined in the Paris agreement. By including degrowth scenarios, excluded in the IPCC reports, the authors claim to make a novel contribution. They construct their scenarios along three dimensions: degree of energy-GDP decoupling, the speed of renewable energy expansion, and the cumulative NETs and CCS deployment. Although the authors make a conceptual difference between “feasibility”, “sustainability”, and “political feasibility”, they should have defined their concept in the introduction, especially the difference between “feasibility” and “political feasibility”. As a point of departure in these concepts, they make the argument that current IPCC scenarios are too optimistic with regards to technological fixes. They also argue that the trust in CCS (carbon capture and storage) and NETs (net emissions technologies) and the possibility for decoupling energy use and GHG emissions from continuous economic growth (measured as GDP) are too optimistic. Technological fixes are not proven on a large scale and the large uncertainties inherent with CCS and NETs, so other mitigation pathways such as degrowth should be given attention.

> We thank you very much for your constructive comments as well as for raising the point of the definition of feasibility. We followed your request and specified our definitions and distinctions within the concept of feasibility in the introduction as follows (lines 91-98):

“We define feasibility, following the IPCC² (p. 52), as ‘the capacity of a system as a whole to achieve a specific outcome’, in our case, a scenario. We additionally distinguish between socio-technical feasibility, broadly following Loftus et al.²³ (i.e. energy-GDP decoupling, speed and scale of the renewable energy transition and NETs deployment), as well as socio-political, including economic, feasibility, broadly following Jewell & Cherp²⁴. The latter define an outcome as politically feasible (p. 2) ‘if there is an agent or group of agents who have the capacity to carry out a set of actions which will lead to that outcome in a given context.’”

Throughout the text we now consistently distinguish between socio-technical feasibility on the one hand and socio-political feasibility on the other.

Their point of departure is the importance of including degrowth scenarios since “not exploring them actually leads to a self-fulfilling prophecy: with research subjectively judging such scenarios as ‘infeasible’ from the start, they remain marginalized in public discourse”. The authors put forward well-known criticism from an ecological economics point of view towards models based on a more orthodox and neoclassical point of view, especially with regards to the possibilities for decoupling energy use from GDP. The authors could have made their epistemological point of view more evident and made it evident that they criticize the more orthodox and neoclassical economic models’ dependence on economic growth and GDP.

> Thank you for pointing this out. We followed your request and made it clearer that we criticise mainstream neoclassical economic approaches. We now specified this where appropriate in our view (new text formatted in italic and underlined; lines 282-286):

“Firstly, Ayres & Warr²⁹, Keen et al.²⁷ and others^{13,26} show that ‘total factor productivity’ (all other production factors influencing economic growth besides capital and labour) is strongly connected to total energy use and its conversion efficiency into useful energy (energy use after accounting for production and conversion losses), contrary to neoclassical economic theory.”

As well as lines 543-544:

“This also implies revisiting the widespread, neoclassical economic optimisation approach in IAMs^{4,22,59}.”

They point on the limitations in the service-based economy from embodied energy and material use as well as outsourcing connected to transferring production of goods to other countries. They

further claim that large-scale renewable energy deployment is unlikely to contribute to material use reduction, since renewables have a considerably higher material footprint than fossil fuels. Their critique centers around the following: 1) Current modelers consider degrowth to be subjectively 'implausible' because GDP is viewed as a suitable indicator of human well-being, it has negative social consequences, and the assumption that it also applies to the global South; 2) The limited representation of behavioral change within integrated assessment modelling compared to the strong supply-side technology focus, thus neglecting demand-side changes that lead to sufficiency; and 3) Some integrated assessment modelling optimizes welfare functions connected to GDP growth and is thus fundamentally unsuited to modelling degrowth.

They propose that the focus should be oriented directly at multidimensional human needs satisfaction. Many degrowth proposals include a strengthening of non-monetary work, such as care work and community engagement, as well as a decommodification of economic activity towards sharing, gifting, and commons. Their proposal also implies a revisiting of the widespread optimization approach in IAMs. The authors claim that this approach implies a plural economic perspective, including along heterodox lines (such as post-Keynesian, ecological, and Marxian economics), to gain a fuller picture of socio-economic reality. Such modelling would also need to broaden the considered portfolio of demand-side measures and behavioral changes to complement the current supply-side technology focus. They also claim that the biophysical foundation of economic activity needs to be considered in much greater detail.

The novelty of the paper is not the arguments per se, which are well-known in other contexts, but in the application of degrowth scenarios to the mitigation of GHG emissions. The paper will be of interest for a broad set of readers interested in pathways for mitigating GHG emissions according to the target set in the Paris agreement. However, the authors do not clarify how to expand the IAMs or the epistemological differences from more orthodox economic thinking. Is it possible to make a radical shift in IAMs and change assumptions, or is it impossible to integrate ecological economic thinking along with neoclassical economics? The authors argue that the expansion in current models must include wider perspectives but do not discuss the model and disciplinary opportunity and barriers for such an expansion.

> We thank you for raising this point. The question of how exactly and how strongly to adapt existing IAMs is a question of considerable complexity, since the models themselves are very complex, heterogeneous and difficult to understand for outsiders. The focus of our paper was therefore to establish the need for such a revision in the first place, which then should include degrowth scenarios. Thus, such modelling would need to take into account several degrowth-related

aspects, such as recognition of biophysical foundations, new wellbeing indicators, demand-side measures and behavioural changes among others. We briefly discussed these aspects to point into potential directions for future research and applications. Further, we name two examples of IAMs which already incorporate several of these aspects, the MEDEAS model and the EUROGREEN model, which could serve as inspiration for the modelling adaptations. Due to the prior comment above, we now make it more explicit that such changes would include revisiting common neoclassical model elements such as optimization based on consumption and GDP, as is also made clear in the cited studies on the mentioned models. Therefore, it is clear that such revisions are a challenge to common neoclassical modelling, albeit a manageable challenge in our opinion, as evidenced by the models and the literature on ecological macroeconomics. However, in our perception it is beyond the scope of the paper to go into more detail on how exactly IAMs would need to change and how this would relate to neoclassical assumptions within the models, especially since this discussion is starting to be raised in the literature and we are very close to the word limit. Thus, we now added only the following sentence to our discussion of possible changes in IAMs regarding degrowth modelling (lines 549-552):

“The necessary detailed discussion of how exactly IAMs would need to change to incorporate some of these features is beyond the scope of this paper, but such discussions are already under way in the literature^{26,61–63} and could be further inspired by current efforts in ecological macroeconomic modelling⁵⁹.”

References 61-63 here are as follows, which already start to discuss changes in our proposed directions:

61. Trutnevyte, E. et al. Societal Transformations in Models for Energy and Climate Policy: The Ambitious Next Step. *One Earth* **1**, 423–433 (2019).
62. Nikas, A. et al. The desirability of transitions in demand: Incorporating behavioural and societal transformations into energy modelling. *Energy Research & Social Science* **70**, 101780 (2020).
63. Pye, S. et al. Modelling net-zero emissions energy systems requires a change in approach. *Climate Policy* **0**, 1–10 (2020).

Treatment of the degrowth scenarios and of the degrowth concept itself was difficult to grasp. Do they mean a sustainable degrowth, implying not recession but planned downscaling, which also considers intra- and intergenerational equity? The authors also do not mention that degrowth

implies different transfer and rebound effects. For example, sufficiency rebound was addressed by Alcott (2010). Individuals and nations who live with less consumption and production create rebound effects that drive consumption and production elsewhere, thus calling for a global degrowth policy and strategy, which also could involve a fairer distribution between countries. Alcott (2010) also argued that the only way to curb rebound effects was to place absolute physical caps on natural resources, such as keeping oil and gas in the ground; if society first caps its resources, people will automatically live more efficiently and sufficiently (Alcott 2014). The authors could have made it more evident what a degrowth policy would mean at a global level, and what transfer effects and unintended effects from such a policy would look like. Here, the author could have expanded their discussion. In that context also reflections from the current COVID-19 situation, on the connection between GDP and GHG- emissions and on how society could be rebuilt post Covid-19 could have been added to their discussion see for example Le Quéré et. Al (2020).

> We thank you for bringing up these points. We followed your request and further specified degrowth in the introduction. We already cite Kallis et al. (2018; Ref. #8) for a definition of degrowth as an “‘equitable downscaling of throughput [that is the energy and resource flows through an economy, strongly coupled to GDP], with a concomitant securing of wellbeing’”, which in our perception already rules out equating degrowth with a recession and emphasizes ‘sustainable’ degrowth which maintains wellbeing. Additionally, we added the following paragraph to make it clearer what degrowth entails, also on a global level (lines 76-85):

“On wellbeing, research^{15,16} shows that high-income countries could scale back their biophysical impact (and GDP), while maintaining (or even increasing^{8,17}) social performance and achieving higher equity among countries. Thus, intra- and intergenerational equity aspects can be taken into account^{8,16,18}, e.g. by making the world economy structurally fairer and redistributing from global North to South^{16,17}. Further, bottom-up studies show that high living standards can be maintained with substantially less per capita energy use than currently consumed in affluent countries¹⁹. However, to ensure that such reductions do not lead to the socially harmful and inequitable effects of a recession requires deep socio-economic changes and policy reforms, such as universal basic services, maximum incomes, working time reductions and democratic firm ownership^{8,16,18,20}.”

Regarding sufficiency rebound effects and Alcott (2010), we further specify caps as suitable degrowth policy (new text formatted in italic and underlined; lines 462-465):

“The crucial question then becomes how, if GDP were to shrink as a result of the required reductions in material and energy use and CO2 (the degrowth hypothesis), e.g. through stringent

eco-taxes *and/or caps*⁸, this GDP decrease could be made socially sustainable, i.e. safeguarding human needs and social function^{8,20}.”

Moreover, to emphasize the discussion on sufficiency rebound effects and the need to take them into account, we now further write in our treatment of socio-political feasibility after the just mentioned sentence and repeating that high living standards can be maintained with much less energy use and GDP (lines 467-471):

“As noted in the introduction, however, substantial socio-economic changes would be necessary to avoid the effects of a recession. Moreover, the reductions and limits would need to be democratically negotiated^{8,20,50} and consider potential ‘sufficiency rebound effects’⁵⁵ (reduced consumption by some being compensated through increases by others), e.g. by international coordination.”

Here, we cite Alcott (2010), as proposed by you, as reference #55.

We also thank you for raising the point of discussing COVID-19. As interesting and fitting as we find this topic to be discussed in relation to degrowth, we think that such a discussion would be beyond the scope of the paper.

The paper is written clearly and in good English. All key assumptions of the paper and how their scenarios are constructed are made evident for the reader and in a way that is possible for others to replicate.

I suggest that the paper could be published in a revised form if the authors make it more evident whether IAMs can expand along their suggested lines, add better definitions of their concepts in the introduction, and provide clearer and broader explanations of degrowth in their context, also including rebound effects and unintended effects from such a policy.

> Thank you very much for your supportive comments and constructive feedback. We hope that the above treatment satisfactorily responded to all your concerns.

REVIEWERS' COMMENTS

Reviewer #1 (Remarks to the Author):

I think the paper is stronger on account of the revisions.

A few minor suggestions (feel free to reject):

It strikes me that there is a slight framing problem that affects the abstract and introduction. In degrowth scholarship, the objective is not to reduce GDP, but to reduce energy and material throughput. What happens to GDP is irrelevant in terms of the ecological outcomes, and (with the right social policies) irrelevant to social outcomes as well. So changing GDP is not an objective as such (even though GDP is likely to decline). But here it seems as though GDP is the variable you're adjusting, which gives rise to language that makes it seem as though the goal of a degrowth scenario is to dial down GDP. I understand why this happens, but it might create some confusion.

Perhaps consider re-writing the first part of the abstract as:

"1.5°C scenarios reported by the Intergovernmental Panel on Climate Change (IPCC) rely on combinations of controversial negative emissions and unprecedented technological change, while assuming continued growth in gross domestic product (GDP). Thus far the IPCC has neglected to consider degrowth scenarios, where rapid emissions reductions are achieved by scaling down aggregate economic output. Thus, their potential to avoid reliance on negative emissions and speculative rates of technological change remains unexplored."

In other words, perhaps at least in this first instance indicate that it is economic output that is being reduced (which is closer to the real objective, namely, reducing energy and material use without relying on speculative decoupling), rather than GDP as such (because GDP is, in the real world, a dependent variable).

Another small note:

In the introduction, you write "Thus, their potential to avoid negative emissions and technological change remains unexplored." I think rewrite as "Thus, their potential to avoid reliance on negative emissions and speculative rates of technological change remains unexplored."

Reviewer #2 (Remarks to the Author):

The authors addressed the comments in my review regarding defining both feasibility and political feasibility more clearly. They distinguish between the socio-technical and socio-political feasibility. However, please check whether sentence 95 and 96 are correct. "as well as socio-political, including economic, feasibility, broadly following Jewell & Cherp", should the correct be economic feasibility in one word?

The authors also expanded their discussion related to degrowth and addressed potential sufficiency rebound in their revision.

The authors clarified their scientific viewpoint regarding their critique of mainstream neoclassical economic approaches, but also stated that current models can be expanded to include "biophysical foundations, new well-being indicators, demand-side measures and behavioral changes among others", and that such changes are "albeit a manageable challenge in our opinion, as evidenced by the models and the literature on ecological macroeconomics". They pointed to literature that shows how other perspectives could expand neoclassical models and IAMs. A shortcoming in that respect is that the authors, of course, did not have time to provide details about the expansion; however, they could have addressed the barriers and opportunities for such a change in modelling beyond the scientific possibility that was described in their paper. This limitation also goes back to the overarching theme of the article, namely the importance of incorporating degrowth scenarios

in mitigation pathways. In other words what are the barriers to expanded models to allow for degrowth scenarios beyond scientific possibilities?

I recommend that the paper be published in its current form, albeit the authors can consider my comments.

Response to Reviewers' Comments

> Our responses are preceded by the > sign as well as formatted in blue colour. We also provide a track-changed version of the revised manuscript additionally to a clean one.

Reviewer #1 (Remarks to the Author):

I think the paper is stronger on account of the revisions.

> We thank you very much for these supportive words and your constructive comments.

A few minor suggestions (feel free to reject):

It strikes me that there is a slight framing problem that affects the abstract and introduction. In degrowth scholarship, the objective is not to reduce GDP, but to reduce energy and material throughput. What happens to GDP is irrelevant in terms of the ecological outcomes, and (with the right social policies) irrelevant to social outcomes as well. So changing GDP is not an objective as such (even though GDP is likely to decline). But here it seems as though GDP is the variable you're adjusting, which gives rise to language that makes it seem as though the goal of a degrowth scenario is to dial down GDP. I understand why this happens, but it might create some confusion.

Perhaps consider re-writing the first part of the abstract as:

"1.5°C scenarios reported by the Intergovernmental Panel on Climate Change (IPCC) rely on combinations of controversial negative emissions and unprecedented technological change, while assuming continued growth in gross domestic product (GDP). Thus far the IPCC has neglected to consider degrowth scenarios, where rapid emissions reductions are achieved by scaling down aggregate economic output. Thus, their potential to avoid reliance on negative emissions and speculative rates of technological change remains unexplored."

In other words, perhaps at least in this first instance indicate that it is economic output that is being reduced (which is closer to the real objective, namely, reducing energy and material use without relying on speculative decoupling), rather than GDP as such (because GDP is, in the real world, a dependent variable).

> Thank you very much for raising this point. We followed your request and changed the abstract in order to avoid the misunderstanding you outlined. The abstract now reads as follows (new text in italic and underlined):

“1.5°C scenarios reported by the Intergovernmental Panel on Climate Change (IPCC) rely on combinations of controversial negative emissions and unprecedented technological change, *while assuming* continued growth in gross domestic product (GDP). *Thus far, the integrated assessment modelling community and the IPCC have neglected to consider degrowth scenarios, where economic output declines due to stringent climate mitigation.* Hence, their potential to avoid reliance on negative emissions and *speculative rates of* technological change remains unexplored. As a first step to address this gap, this paper compares 1.5°C degrowth scenarios with IPCC archetype scenarios, using a simplified quantitative representation of the fuel-energy-emissions nexus. We find that the degrowth scenarios minimize many risks for feasibility and sustainability compared to technology-driven pathways, such as the reliance on high energy-GDP decoupling, large-scale carbon dioxide removal and large-scale and high-speed renewable energy transformation. However, substantial challenges remain regarding political feasibility. *Nevertheless, [text removed]* degrowth pathways should be thoroughly considered.”

We also address the integrated assessment modelling (IAM) community, since the IPCC only summarizes existing modelling and thus the IAM community is crucial for changes in the IPCC as well. We further stuck with our formulation “...due to stringent climate mitigation” to emphasize that in a degrowth scenario the objective is still to reduce carbon emissions, but the result is most likely a reduction in economic output and GDP. Due to the changes we needed to remove some words as is done in the last sentence.

Another small note:

In the introduction, you write "Thus, their potential to avoid negative emissions and technological change remains unexplored." I think rewrite as "Thus, their potential to avoid reliance on negative emissions and speculative rates of technological change remains unexplored."

> Thank you very much for highlighting this. We followed your request and inserted the specification, as shown above.

Reviewer #2 (Remarks to the Author):

The authors addressed the comments in my review regarding defining both feasibility and political feasibility more clearly. They distinguish between the socio-technical and socio-political feasibility. However, please check whether sentence 95 and 96 are correct. “as well as socio-political, including economic, feasibility, broadly following Jewell & Cherp”, should the correct be economic feasibility in one word?

The authors also expanded their discussion related to degrowth and addressed potential sufficiency rebound in their revision.

> We thank you very much for your constructive comments. We followed your request and changed the sentence in lines 95-96 as follows:

“...as well as socio-political feasibility, which includes economic feasibility, broadly following Jewell & Cherp”

The authors clarified their scientific viewpoint regarding their critique of mainstream neoclassical economic approaches, but also stated that current models can be expanded to include “biophysical foundations, new well-being indicators, demand-side measures and behavioral changes among others”, and that such changes are “albeit a manageable challenge in our opinion, as evidenced by the models and the literature on ecological macroeconomics”. They pointed to literature that shows how other perspectives could expand neoclassical models and IAMs. A shortcoming in that respect is that the authors, of course, did not have time to provide details about the expansion; however, they could have addressed the barriers and opportunities for such a change in modelling beyond the scientific possibility that was described in their paper. This limitation also goes back to the overarching theme of the article, namely the importance of incorporating degrowth scenarios in mitigation pathways. In other words what are the barriers to expanded models to allow for degrowth scenarios beyond scientific possibilities?

I recommend that the paper be published in its current form, albeit the authors can consider my comments.

> Thank you very much for highlighting this and further explaining your comment. We are aware that there are further barriers beyond sole technical modelling possibilities to changing current modelling practices such that they can include degrowth scenarios, e.g. the believe of modellers that economic growth is necessary for sustaining human wellbeing or more extreme, that it equals

human progress. We now shortly address such views in the introduction as well as in the section on economic feasibility. However, we also see that a fair treatment of the complexity of the issue, e.g. considering worldviews of modellers and their philosophical assumptions, funding structures, mainstream economic citation networks, power structures in the IPCC etc., would be beyond the scope of this paper, but we hope that our work might encourage others to focus more on non-technical barriers to degrowth modelling. We therefore decided not to change the manuscript further, but very much appreciate your comments.